# Universal Rates for Active Learning

**Steve Hanneke**
Purdue University
steve.hanneke@gmail.com

**Amin Karbasi**
Yale University
amin.karbasi@yale.edu

**Shay Moran**
Technion, Google Research
smoran@technion.ac.il

**Grigoris Velegkas**
Yale University
grigoris.velegkas@yale.edu

## Abstract

In this work we study the problem of *actively* learning binary classifiers from a given concept class, i.e., learning by utilizing *unlabeled* data and submitting targeted *queries* about their labels to a domain expert. We evaluate the quality of our solutions by considering the *learning curves* they induce, i.e., the rate of decrease of the misclassification probability as the number of label queries increases. The majority of the literature on active learning has focused on obtaining *uniform* guarantees on the error rate which are only able to explain the upper envelope of the learning curves over families of different data-generating distributions. We diverge from this line of work and we focus on the *distribution-dependent* framework of universal learning whose goal is to obtain guarantees that hold for any fixed distribution, but do not apply uniformly over all the distributions. We provide a complete characterization of the optimal learning rates that are achievable by algorithms that have to specify the number of *unlabeled* examples they use ahead of their execution. Moreover, we identify combinatorial complexity measures that give rise to each case of our tetrachotomic characterization. This resolves an open question that was posed by Balcan et al. (2010). As a byproduct of our main result, we develop an active learning algorithm for *partial* concept classes that achieves exponential learning rates in the uniform setting.

## 1 Introduction

The most prototypical type of learning is that of *supervised* learning, where an algorithm is given as input $n$ *labeled* data points sampled i.i.d. from some unknown distribution and the goal is to output a function that has low probability of misclassifying new data from the same distribution. One caveat with this passive model of learning is that it fails to capture the settings in which *unlabeled* data is easily accessible, but obtaining their labels is costly. A natural example that fits this description is that of webpage classification. It is easy for any web crawler to collect information about billions of different webpages in a very short amount of time, however understanding which category a webpage belongs to usually requires human feedback. Perhaps the most commonly used way to model this problem is through the *active* learning framework in which the learner is given access to a large stream of unlabeled data and a budget of $n$ queries that it can submit to a domain expert in order to obtain the label of some datapoint. The learner's goal is to submit queries for the labels of the most informative examples, and, as a result, eliminate some redundancy in the information content of labeled data.

In this paper we study the optimal learning rates that are achievable by an active learning algorithm. In the passive learning setting, it is common to measure the quality of an algorithm by its *learning curve*, i.e., plotting the decay of its error rate as the number of training examples increases. In contrast, in the

38th Conference on Neural Information Processing Systems (NeurIPS 2024).

active learning setting, the resource we are interested in is the number of label queries the algorithm requests, so it is natural to consider the rate of decay of the misclassification probability as the number of queries $n$ increases. Most of the prior works on active learning have focused on obtaining *uniform* guarantees on the error rate, i.e., guarantees that hold uniformly over the data-generating distributions. In this work we follow a different path and we view the problem through the lens of *universal* rates that was recently introduced by Bousquet et al. (2021). In this framework we are aiming for guarantees that hold for all distributions, but they do not hold uniformly over them. In other words, we allow for *distribution-dependent* constants in the error rate. To make the distinction between uniform and universal rates clear, we first recall what uniform learnability of a hypothesis class $\mathbb{H}$ means. We say that $\mathbb{H}$ is uniformly learnable at rate $R(n)$ if there exists a learning rule $\hat{h}_n$ such that

$$\left(\exists C, c > 0\right)\left(\forall P \in \mathrm{RE}(\mathbb{H})\right) \text{ it holds that } \mathbb{E}[\mathrm{er}_P(\hat{h}_n)] \leq C \cdot R(c \cdot n), \forall n \in \mathbb{N}.$$

The above expression states that there exists a learning rule $\hat{h}_n$ and *distribution-independent* constants such that for all realizable distributions the expected error rate of the classifier is at most $C \cdot R(c \cdot n)$. The difference in the definition of universal learnability is that we swap the order of the quantifiers. To be more precise, we say that a class $\mathbb{H}$ is universally learnable at rate $R(n)$ if there exists a learning rule $\hat{h}_n$ such that

$$\left(\forall P \in \mathrm{RE}(\mathbb{H})\right)\left(\exists C, c > 0\right) \text{ such that } \mathbb{E}[\mathrm{er}_P(\hat{h}_n)] \leq C \cdot R(c \cdot n), \forall n \in \mathbb{N}.$$

Note that in the above definition the constants $c, C$ are *distribution-dependent.* As is evident from our main result and from prior results in universal learning (Bousquet et al., 2021; Hanneke et al., 2022; Kalavasis et al., 2022; Bousquet et al., 2022; Hanneke et al., 2023) this change in the definition affects the landscape of the optimal learning rates significantly.

## 1.1 Related Work

**Active Learning.** There has been a very long line of work deriving theoretical guarantees for active learning, both in the realizable and the agnostic setting (Cohn et al., 1994; Dasgupta, 2005; Balcan et al., 2006; Hanneke, 2007b,a; Dasgupta et al., 2007; Hanneke, 2009; Balcan et al., 2009, 2010; Hanneke, 2012, 2014; Wiener et al., 2015; Hanneke and Yang, 2015; Beygelzimer et al., 2016). As we alluded to before, most of these works focus on obtaining *minmax* guarantees, i.e., the bounds they provide hold in the worst case over a family of distributions. To be more precise, while many of these results have distribution-dependent guarantees, they are typically expressed in a way that aims to match a lower bound on the minmax performance over a family o distributions, with respect to some parameter, like the disagreement coefficient. For these reasons the results in these works do not capture the full spectrum of universal rates. We also remark that there are a few works that do study universal rates, e.g. Balcan et al. (2010); Hanneke (2012); Yang and Hanneke (2013), but none of them have derived a complete characterization of the optimal rates.

**Universal Rates.** The study of universal learning rates was put forth in the seminal work of Bousquet et al. (2021) who derived a complete characterization of the optimal rates in the supervised learning setting. Later, Kalavasis et al. (2022) extended these result to the multiclass setting, with a bounded number of classes, and Hanneke et al. (2023) improved upon this result by characterizing multiclass classification with an infinite number of labels. Subsequently, the work of Bousquet et al. (2022) derived more fine-grained results for binary classification compared to Bousquet et al. (2021). The work that is most closely related to ours is Hanneke et al. (2022) which derives a complete characterization of the optimal learning rates in a very general interactive learning setting. In that setting, the learner is allowed to submit arbitrary binary valued queries about the unlabeled data. We provide a detailed comparison between our results and theirs in Section 1.5. Very recently, Attias et al. (2024) studied universal rates in the context of regression.

**Partial Concept Classes.** The vast majority of the literature in learning theory has focused on *total* concept classes, i.e., classes that consist of functions that are defined everywhere on the instance domain. Recently, Alon et al. (2021) proposed a learning theory of *partial* concept classes, i.e., classes that consist of functions that can be *undefined* on some parts of the instance domain. Later, Kalavasis et al. (2022) extended some of the results to the multiclass setting, with a finite number of labels. Recently, Cheung et al. (2023) studied partial concept classes in the context of online learning and

they showed that there are such classes that are online learnable but none of their "extensions" to total concept classes is online learnable. The advantage of partial concepts is that they provide a convenient way to express *data-dependent* constraints. En route of obtaining our main result, we design active learning algorithms for partial concept classes. For a more detailed discussion about partial concepts we refer the reader to Appendix A.2.

## 1.2 Formal Setting

**Learning Model.** We now present formally the learning setting that we consider in this work. There is a *domain* $\mathcal{X}$, which we assume to be a Polish space, and a concept class $\mathbb{H} \subseteq \{0, 1\}^{\mathcal{X}}$, which satisfies standard measurability assumptions (see Definition A.1). We define a *classifier* $h : \mathcal{X} \rightarrow \{0, 1\}$ to be a universally measurable function and its *error rate* is defined as $\mathrm{er}_\mathrm{P}(h) := \mathrm{P}\left[(x, y) : h(x) \neq y\right]$, where P is the data-generating distribution over $\mathcal{X} \times \{0, 1\}$. When P is clear from the context we might drop the subscript P in $\mathrm{er}_\mathrm{P}(h)$. We call P *realizable* with respect to the hypothesis class $\mathbb{H}$ if $\inf_{h \in \mathbb{H}} \mathrm{er}_\mathrm{P}(h) = 0$. We denote by $\mathrm{P}_\mathcal{X}$ be the marginal distribution of P on $\mathcal{X}$.

**Active Learning Model.** We define an *active learning algorithm* to be a sequence of universally measurable functions which, given access to a stream of *unlabeled* data points from $\mathcal{X}$ that are drawn i.i.d. from $\mathrm{P}_\mathcal{X}$ and a label query budget $n$, output a classifier $\hat{h}_n : \mathcal{X} \times \{0, 1\}$. In this work we consider a *non-adaptive* active learning model, with respect to the *unlabeled* data. In particular, the learning algorithm needs to specify a function $u : \mathbb{N} \rightarrow \mathbb{N}$ so that $u(n)$ is the number of *unlabeled* points it observes and $n$ is the number of points for which it can request the label. The number $u(n)$ is specified *before* the execution of the algorithm and cannot be modified based on the realization of the unlabeled sequence or the answers that it gets for the labels of the points it queries. We place no bound whatsoever on the function $u(\cdot)$. Also, the algorithm can only request the labels of points it has observed and not arbitrary points from $\mathcal{X}$. We emphasize that the label requests that the algorithm makes can be adaptive, and can depend on answers to previous label queries. To the best of our knowledge, the active learning algorithms that have been proposed in the literature either fit into this model or they can be modified to satisfy the non-adaptivity restriction with negligible performance loss.

**Learning Rates.** We now define formally what it means for an algorithm to achieve a learning rate $R(n)$ in the *universal learning* model. We adopt the definition of Bousquet et al. (2021).

**Definition 1.1** (Learning Rates (Bousquet et al., 2021)). *Fix a concept class $\mathbb{H}$, and let $R : \mathbb{N} \rightarrow [0, 1], R(n) \overset{n \rightarrow \infty}{\longrightarrow} 0$ be a rate function, where $n$ is the label query budget of the learner.*

- *$\mathbb{H}$ is learnable at rate $R$ if there is a learning algorithm $\hat{h}_n$ such that for every realizable distribution P, there exist $c, C$ for which $\mathbb{E}[\mathrm{er}(\hat{h}_n)] \leq CR(cn), \forall n \in \mathbb{N}$.*

- *$\mathbb{H}$ is not learnable at rate faster than $R$ if for all learning algorithms $\hat{h}_n$ there exists a realizable distribution P and $c, C$ for which $\mathbb{E}[\mathrm{er}(\hat{h}_n)] \geq CR(cn)$, for infinitely many $n \in \mathbb{N}$.*

- *$\mathbb{H}$ is learnable with optimal rate $R$ if it is learnable at rate $R$ and it is not learnable at rate faster than $R$.*

- *$\mathbb{H}$ admits arbitrarily fast rates if for all rate functions $R$, it is learnable at rate $R$.*

- *$\mathbb{H}$ requires arbitrarily slow rates if for all rate functions $R$, it is not learnable at rate faster than $R$.*

**Combinatorial Measures.** We now define some combinatorial complexity measures that our characterization relies on. To make the presentation easier to follow, we provide informal definitions. For the formal ones, we refer the reader to Appendix A.3. We first describe the *Littlestone tree* that was introduced by Bousquet et al. (2021).

**Definition 1.2** (Littlestone Tree, Informal (see Definition A.8) (Bousquet et al., 2021)). *A Littlestone tree for $\mathbb{H} \subseteq \{0, 1\}^{\mathcal{X}}$ is a complete binary tree of depth $d \leq \infty$ whose nodes are labeled by elements of $\mathcal{X}$ and the edges to the left, right child are labeled by $0, 1$. We require that for every level $0 \leq n < d$*

*and every path from the root to a node at level $n$ there is some $h \in \mathbb{H}$ that realizes this path. We say that $\mathbb{H}$ has an infinite Littlestone tree if it has a Littlestone tree of depth $d = \infty$.*

We underline that this notion can be thought of as an infinite extension of the *Littlestone dimension* (Littlestone, 1988) of $\mathbb{H}$. Recall that the Littlestone dimension is defined to be the largest $d \in \mathbb{N}$ for which $\mathbb{H}$ has a Littlestone tree of such depth and it is $\infty$ if one can construct Littlestone trees of arbitrary depth. Crucially, this is not the same as having a *single* tree whose depth is infinite, so one can see that infinite Littlestone dimension is *not* the same as having an infinite Littlestone tree.

We next give the definition of the *Vapnik-Chervonenkis-Littlestone tree* (VCL) that was introduced by Bousquet et al. (2021).

**Definition 1.3** (VCL Tree, Informal (see Definition A.10) (Bousquet et al., 2021)). *A VCL tree for $\mathbb{H} \subseteq \{0,1\}^{\mathcal{X}}$ is a complete tree of depth $d \leq \infty$ such that every level $0 \leq n < d$ has nodes that are labeled by $\mathcal{X}^{n+1}$ with branching factor $2^{n+1}$ and whose $2^{n+1}$ edges connecting a node to its children are labeled by the elements of $\{0,1\}^{n+1}$. We require that for every node at any level $0 \leq n < d$, the path from the root to this node is realized by some $h \in \mathbb{H}$. We say that $\mathbb{H}$ has an infinite VCL tree if it has a VCL tree of depth $d = \infty$.*

Intuitively, the VCL tree combines the notions of the Littlestone tree and the VC dimension (Vapnik and Chervonenkis, 1971; Blumer et al., 1989). The differences between the Littlestone tree and the VCL tree are that in the latter the size of the nodes increases linearly with the level and the branching factor increases exponentially, whereas in the former all the nodes are singletons and the branching factor is always two.

We are now ready to introduce a new combinatorial measure which we call the *star tree*.

**Definition 1.4** (Star Tree, Informal (see Definition A.9)). *A star tree for $\mathbb{H} \subseteq \{0,1\}^{\mathcal{X}}$ is a complete tree of depth $d \leq \infty$ such that every level $0 \leq n < d$ has nodes that are labeled by $(\mathcal{X} \times \{0,1\})^{n+1}$ with branching factor $n+1$ and whose $n+1$ edges connecting a node to its children are labeled by $\{0, \dots, n\}$. The label of the edge indicates the element of the node whose label along the path is flipped. We require that for every node at level $0 \leq n < d$, the path from the root to this node is realized by $\mathbb{H}$. We say that $\mathbb{H}$ has an infinite star star tree if it has a star tree of depth $d = \infty$.*

Essentially, this definition combines the structure of a Littlestone tree with the notion of the *star number* (Hanneke and Yang, 2015). Every node at level $n$ consists of $n+1$ *labeled* points and there are $n+1$ edges attached to it. Each edge indicates which of the $n+1$ points of the labeled node has its label flipped along every path that this edge is part of.

## 1.3 Main Results

We are now ready to state the main results of this work. Our first main result is a complete characterization of the optimal learning rates that a class $\mathbb{H}$ admits in the active learning setting.

**Theorem 1.5.** *For every concept class $\mathbb{H}$ exactly one of the following cases holds.*

- $\mathbb{H}$ *is actively learnable at arbitrarily fast rates.*

- $\mathbb{H}$ *is actively learnable at an optimal rate $e^{-n}$.*

- $\mathbb{H}$ *is actively learnable at $o(1/n)$ rates, but requires rates arbitrarily close to $1/n$.*

- $\mathbb{H}$ *requires arbitrarily slow rates for active learning.*

Our next result characterizes exactly when these rates occur by specifying combinatorial complexity measures of $\mathbb{H}$ that determine which case of the tetrachotomy it falls into.

**Theorem 1.6.** *For every concept class $\mathbb{H}$ the following hold.*

- *If $\mathbb{H}$ does not have an infinite Littlestone tree, then it is learnable at arbitrarily fast rates.*

- *If $\mathbb{H}$ has an infinite Littlestone tree but does not have an infinite star tree, then it is learnable at optimal rate $e^{-n}$.*

- *If $\mathbb{H}$ has an infinite star tree but does not have an infinite VCL tree, then it is learnable at optimal rate $o(1/n)$, but requires rates arbitrarily close to $1/n$.*

- *If $\mathbb{H}$ has an infinite VCL tree, then it requires arbitrarily slow rates.*

We remark that the landscape of the optimal rates looks significantly different compared to the passive setting (Bousquet et al., 2021) and the general interactive learning setting (Hanneke et al., 2022). Our main result answers an open question that was posed in Balcan et al. (2010), which asks for necessary and sufficient conditions for learnability at an exponential rate in the active learning setting.

## 1.4 Examples

We now present examples of classes that witness each case of our tetrachotomic characterization.

Arbitrarily-fast rates: Bousquet et al. (2021) give examples with no infinite Littlestone tree (e.g., thresholds on the integers, and positive halfspaces on $\mathbb{N}^d$), hence learnable at arbitrarily fast rates.

Exponential rates: Famously, threshold classifiers $[a, \infty)$ over $\mathbb{R}$ have an infinite Littlestone tree and are actively learnable at exponential rate (even uniformly). A more interesting example is the class of interval classifiers $[a, b]$ on $\mathbb{R}$, which (also famously) has uniform rate $1/n$ for active learning (it has infinite star number), and has an infinite Littlestone tree, but it *has no infinite star tree*: for any choice of $x$ in the root node, the edge labeling $x$ as 1 has a version space with star number equal 4 (it is effectively 2 disjoint threshold problems), so the depth of that subtree is bounded. Therefore, by our theory, it is actively learnable at $e^{-n}$ universal rate (in stark contrast to the uniform rate $1/n$).

Sublinear rates: Halfspaces on $\mathbb{R}^d$ have an infinite star tree but no infinite VCL tree. To construct that star tree, the key observation is that any set in convex position is a star set (Balcan et al., 2010).

Arbitrarily slow rates: Bousquet et al. (2021) provide examples of classes with an infinite VCL tree, such as the class of all claffisifiers, so these require arbitrarily slow rates in our setting as well.

## 1.5 Comparison to Supervised Learning and General Interactive Learning Setting

We now compare our characterization to the results of Bousquet et al. (2021) that prove an analogous result in the supervised learning setting. Let us first recall their main result which shows that in this learning model a class is universally learnable at:

- Exponential rate $e^{-n}$ if and only if it does not have an infinite Littlestone tree.
- Linear rate $1/n$ if and only if it has an infinite Littlestone tree but does not have an infinite VCL tree.
- Arbitrarily slow rates if and only if it has an infinite VCL tree.

As one can see, our results illustrate the advantage that active learning algorithms have over their supervised counterparts. We underline that the only case where active learning does not offer an improvement compared to the passive setting is when $\mathbb{H}$ has an infinite VCL tree.

Let us now discuss the interactive learning model that was considered by Hanneke et al. (2022). In this general model of interaction, the learner is allowed to ask *arbitrary binary queries* about any subset of the unlabeled data. In particular, these queries include, but are not limited to, label queries, comparison queries, and general membership queries. They show that the optimal rates that any class $\mathbb{H}$ admits are the following:

- Arbitrarily fast if and only if it does not have an infinite Littlestone tree.
- Exponential if and only if it has an infinite Littlestone tree but not an infinite VCL tree.
- Arbitrarily slow if and only if it has an infinite VCL tree.

We underline that the algorithm by Hanneke et al. (2022) that achieves arbitrarily fast rates uses only label queries and the number of unlabeled data $u(n)$ that it uses can be chosen statically prior to the execution of the algorithm. Therefore, this result applies to the setting we consider in our work. Moreover, the lower bounds they provide also apply to our setting since (i) the queries that Hanneke et al. (2022) consider are more general, and (ii) their lower bounds hold even when the learner knows the marginal distribution $P_{\mathcal{X}}$. As is evident from our main result, the active learning setting provides a richer landscape of optimal rates compared to the general interactive learning setting of Hanneke et al. (2022). In particular, just the absence of an infinite VCL tree for $\mathbb{H}$ does not necessarily imply

that we can achieve (at least) exponential rates. To get this result, Hanneke et al. (2022) make strong use of these general queries in order to provide an algorithm that has a binary-search flavor and can learn *partial* concept classes with finite VC dimension at an exponential rate. Our main result shows that such guarantees are not achievable by using only label queries.

### 1.6 Technical Challenges

The most technically innovative part of our work is the $o(1/n)$ algorithm. Let us set up some terminology to facilitate our discussion. The *version space* of a concept class, given a labeled dataset $S$, is the set of concepts that classify all the elements of $S$ correctly. Moreover, the *VCL game* is a Gale-Stewart game defined by Bousquet et al. (2021) that was used in their passive learning algorithm that achieves $1/n$ universal rate. The original (passive) learner from Bousquet et al. (2021) partitions a portion of the data into some number $B(n)$ of batches, from which they construct partial concept classes of finite VC dimension by using the batches to play the VCL game against the player's winning strategy, and for each resulting partial concept class they run a separate learner on another portion of data and return a majority vote of their resulting classifiers. The latter part of this strategy *could never work* in the active learning setting, since the number of batches $B(n)$ must be an increasing function of $n$, and applying an active learner for each partial concept class separately would require each to use (nearly) $\Omega(n)$ queries (to get the $o(1/n)$ guarantee there), so the total number of queries would be (nearly) $\Omega(B(n)n) \gg n$, violating the label budget $n$. To resolve this, we ended up completely re-imagining how to use these partial concept classes. Rather than running a separate algorithm for each class, we run a *single* active learning algorithm, where individual decisions of whether to query involve voting over the partial concept classes. We first extend an algorithm of Hanneke (2012) (for total concepts) to achieve $o(1/n)$ rate for partial concept VC classes (this required completely re-formulating Hanneke's analysis). The resulting algorithm involves estimating the probability that a random $k$-tuple is VC shattered by certain constrained version spaces. We replace this with an estimated average (over a select subset of partial concept classes) of this shattering probability, which we show composes appropriately with the analysis of the algorithm to obtain the $o(1/n)$ rate. Comparing our work to the general interactive learning setting considered by Hanneke et al. (2022), the main differences are (i) we design a new algorithm that uses only label queries and achieves exponential rates when $\mathbb{H}$ does not have an infinite star tree, a combinatorial measure we introduce in our work, (ii) we prove a $o(1/n)$ lower bound in the setting where $\mathbb{H}$ has an infinite star tree, and (iii) we propose a novel active learning algorithm that achieves sublinear rates when $\mathbb{H}$ does not have an infinite VCL tree.

## 2 Arbitrarily Fast Rates

As we explained before, the first case of the tetrachotomy in our characterization is a direct implication of the results by Hanneke et al. (2022). To be more precise, they design an algorithm which achieves arbitrarily fast rates using only label queries. In order to do that, they need the number of *unlabeled* points $u(n)$ to be an arbitrarily fast increasing function, that, nevertheless, can be specified in a non-adaptive manner prior to the execution of the algorithm. The result is summarized in Theorem 2.1.

**Theorem 2.1** (Hanneke et al. (2022)). *If $\mathbb{H}$ does not have an infinite Littlestone tree it is actively learnable with arbitrarily fast rates.*

For completeness, we present their algorithm in Figure 1. Similary with the upper bound, the lower bound is an immediate consequence of a result from Hanneke et al. (2022), since the learner can submit more informative queries in their model.

**Theorem 2.2** (Hanneke et al. (2022)). *If $\mathbb{H}$ has an infinite Littlestone tree, then $\mathbb{H}$ is not actively learnable at rate faster than exponential $e^{-n}$. This holds even if $P_{\mathcal{X}}$ is known to the learner.*

## 3 Exponentially Fast Rates

In this section, we prove the second case in the tetrachotomy we have stated, i.e., that $\mathbb{H}$ is learnable at an exponential rate if and only if it has an infinite Littlestone tree and it does not have an infinite star tree. Our proof consists of two parts. First, we show that if $\mathbb{H}$ does not have an infinite star tree it is learnable at an exponentially fast rate. Then, we show that whenever $\mathbb{H}$ has an infinite star tree, the best achievable rate cannot exceed $o(1/n)$. The omitted details can be found in Appendix C.

### 3.1 Exponential Rates Algorithm: High-Level Overview

The high-level approach to get the exponential rates algorithm follows the same spirit of the approaches from Bousquet et al. (2021); Hanneke et al. (2022). First, we design an appropriate Gale-Stewart game (cf. Appendix A.2), i.e., a game between a learner and an adversary, that is associated with $\mathbb{H}$ in which the learner has a winning strategy if and only if $\mathbb{H}$ does not have an infinite star tree. The next step is to show that, in the limit, the winning strategy of the learner gives rise to a *partial* concept class $\mathcal{F}$ that has *finite* star number (see Definition A.7). Since this result is asymptotic, our approach is to consider several instances of this game, execute them for a finite number of steps, and obtain a partial concept class from each one. Then, we aggregate these classes into a *majority* class. The intuition is that, with high probability, most of the games will have induced classes whose star number is bounded by a distribution-dependent constant, so then we can show that the majority class will also have bounded star number, and this bound is distribution-dependent. Thus, our task boils down to actively learning a partial concept class whose star number is finite. In order to do that, we extend the approach that Hanneke and Yang (2015) used for total concept classes to the regime of partial classes. We believe that this result could be of independent interest.

### 3.2 The Star Tree Game

We first outline the Gale-Stewart game we use. Recall that every node of a star tree at depth $n$ consists of $n + 1$ points along with their labels. There are $n + 1$ edges that connect the node with its children and the label of every edge indicates the point whose label is flipped along any path that uses this edge. Let us now describe the game $\mathfrak{G}$ that we use in this setting. In every round $\tau \geq 1$ we have the following interaction between the learner $P_L$ and the adversary $P_A$:

- Player $P_A$ chooses points $(\vec{\xi}_\tau, \vec{\zeta}_\tau) = (\xi_\tau^0, \ldots, \xi_\tau^{\tau-1}, \zeta_\tau^0, \ldots, \zeta_\tau^{\tau-1}) \in \mathcal{X}^\tau \times \{0,1\}^\tau$.
- Player $P_L$ chooses $\eta_\tau \in \{0, \ldots, \tau - 1\}$.
- Player $P_L$ wins the game in round $\tau$ if

$$\mathbb{H}_{\vec{\xi}_1, \vec{\zeta}_1, \eta_1, \ldots, \vec{\xi}_\tau, \vec{\zeta}_\tau, \eta_\tau} := \left\{ h \in \mathbb{H} : \begin{array}{ll} h(\xi_s^i) = \zeta_s^i, & \text{if } \eta_s \neq i \\ h(\xi_s^i) = 1 - \zeta_s^i, & \text{if } \eta_s = i \end{array}, 1 \leq s \leq \tau, 0 \leq i < s \right\} = \emptyset. \tag{1}$$

It is easy to see that the winning condition for $P_L$ is finitely decidable (cf. Appendix A.2), hence $\mathfrak{G}$ is a Gale-Stewart game. Recall that this means exactly one player between the adversary and the learner has a winning strategy. Using a result regarding the measurability of winning strategies in Gale-Stewart games that was shown by Bousquet et al. (2021) (see Theorem A.3) we can prove the following connection between $\mathfrak{G}$ and the existence of infinite star trees ( see Appendix C.1 for the proof).

**Lemma 3.1.** *The class $\mathbb{H}$ does not have an infinite star tree if and only if $P_L$ has a universally measurable winning strategy in $\mathfrak{G}$.*

The first step in our approach, is to make use of some $\Theta(n)$ label queries in order to obtain the labels of $\Theta(n)$ many points. The idea these labeled points in order to simulate the Gale-Stewart game we described above (see Appendix C.3 for the details). The main technical issue we need to handle is that we have no control over the number of rounds the game needs in order to terminate. Using ideas that have appeared in the universal learning literature, we use a portion of these labeled points to estimate some number $\hat{t}_n$ so that, with at least some constant probability over the dataset, the game will terminate within $\hat{t}_n$ many rounds. Then, we split the remaining of the labeled dataset into batches of size $\hat{t}_n$ and we run the game on each batch. The outcome of each game gives us a *pattern-avoidance* function, i.e., a function that takes an input tuples of arbitrarily *labeled* points and changes the label of one of them so that the resulting labeled tuple is not consistent with the data-generating distribution $P$. In other words, the output of this function could not have been generated by the $P$. A technical complication we need to handle is that we obtain multiple such pattern avoidance functions, some of which are incorrect, but to make the presentation cleaner we explain the idea using a single pattern avoidance function $\widetilde{g}_{t^*}$ that produces inconsistent labels. The formal setting is handled in Appendix C.4. One way to think about this function is that it provides *data-dependent* constraints. Thus, it is natural to express such a constraint through a *partial* concept class. We define

$$\mathcal{F} = \left\{ f : \mathcal{X} \to \{0, 1, \star\} : (x_1, f(x_1)), (x_2, f(x_2)), \ldots, (x_{t^*}, f(x_{t^*})) \notin \text{image}(\widetilde{g}_{t^*}), \forall x_1, \ldots, x_{t^*} \in \mathcal{X}^{t^*} \right\}.$$

Notice that the constraint we have placed on $\mathcal{F}$ is satisfied if $f(x_i) = \star$, for some $i \in [t^*]$. We consider the natural extension of the notion of star sets to the case of partial concept classes, i.e., we say that a labeled set $S$ with labels in $\{0, 1\}$ is a star set if $S$ and its adjacent sets $S'$, whose labels are still restricted to be in $\{0, 1\}$, are obtainable using functions from $\mathcal{F}$ (see Definition A.7). A key observation is that the star number of $\mathcal{F}$ is bounded by $t^* - 1$. Another difficulty we need to overcome is that we do not know $t^*$, since it is a random variable that depends on the realized sequence and its distribution might have heavy tails. To make our approach easier to follow, let us first assume that we do know $t^{*1}$. Then, our task boils down to actively learning a partial concept class.

### 3.3 Active Learning of Partial Concept Classes with Finite Star Number

We now present an algorithm that achieves exponential rates when actively learning a partial concept class $\mathcal{F}$ that star number $\mathfrak{s} < \infty$. The idea of our approach is to reduce the problem of actively learning a partial concept to the well-studied problem of actively learning a total concept class. The algorithm is presented in Figure 2. Let us explain the high-level ideas of the algorithm. First, we consider a large enough set of unlabeled data. Our goal is to find their labels using *logarithmically* many queries. To do that, we consider the uniform distribution over these unlabeled examples. Then, we use an algorithm from Hanneke and Yang (2015) (see Theorem C.1) which guarantees exponential rates when applied to a class with finite star number. Because the underlying distribution on the sample is uniform, with high probability, the algorithm will find the correct labels of all the points. Finally, we feed these labeled examples to the one-inclusion graph algorithm (Theorem A.5) to get the desired result. We are now ready to state our theorem. The proof is postponed to Appendix C.2.

**Theorem 3.2.** *There exists an active learning algorithm $\mathcal{A}$ for a partial concept class $\mathcal{F}$ which given a label budget $n$ and access to unlabeled samples from a realizable distribution $\mathrm{P}^*$ returns a classifier $\hat{h}_n$ such that $\mathbb{E}_{\mathrm{P}^*}[\mathrm{er}(\hat{h}_n)] \leq c_1 \mathrm{d} e^{-c_2 n/\mathfrak{s}}$, where $c_1, c_2$ are absolute numerical constants, and $\mathfrak{s}, \mathrm{d}$ is the star number, VC dimension of $\mathcal{F}$.*

### 3.4 Slower than Exponential is Sublinear

The next step in the characterization is to show that if $\mathbb{H}$ has an infinite star tree, then it does not admit rates faster than sublinear. The proof starts by picking a random path on the infinite star tree. The target distribution is supported only on nodes of the selected path. Then, given some algorithm $\hat{h}_n$, we distribute the mass of the target probability distribution across the path, potentially skipping some nodes of it, in a way that creates an infinite sequence $n_{i_1}, n_{i_2}, \ldots$, so that when the learner has label budget $n_{i_j}$, with some constant probability, it will only observe unlabeled points up to level $k_{i_j}$. Moreover, with at least some constant probability, it will not query the point of that level whose label is flipped along the target path. On that event, it makes a mistake with probability at least $C \cdot p_{i_j}/k_{i_j}$, where $C$ is some absolute constant. Our choice of $p_{i_j}, k_{i_j}$ guarantees that $R(n_j) > p_{i_j}/k_{i_j}$, where $R(\cdot)$ is the target sublinear rate function. Finally, we apply Fatou's lemma to get the desired result. For the full proof and the formal theorem statement, we refer the reader to Appendix C.5

## 4 Sublinear Rates

Our approach to achieve sublinear rates in the setting where $\mathbb{H}$ does not have an infinite VCL tree shares some high-level ideas with the one in Section 3, but many technical challenges make it significantly more involved. The main obstacle is that there is no active learning algorithm that achieves sublinear rates for VC classes *uniformly* over all realizable distributions. Recall that in the exponential rates setting, such an algorithm does exist (Hanneke and Yang, 2015). Instead, the sublinear rates algorithm for VC classes from Hanneke (2012) depends on distribution-dependent constants in the sample complexity. The omitted details from this section can be found in Appendix D.

Instead of the star tree Gale-Stewart game that was used to get the exponential rates guarantee, we use the VCL Gale-Stewart game Bousquet et al. (2021) (cf. Figure 6). To be more precise, we use $\lfloor n/5 \rfloor$ of the label budget to get the labels of $\lfloor n/5 \rfloor$ unlabeled points that come i.i.d. from $\mathrm{P}_\mathcal{X}$. Then, we execute the VCL game on $\Theta(\sqrt{n})$ different batches of size $\Theta(\sqrt{n})$. Each of these games induces a *partial* concept class. We show how to obtain a P-dependent bound on the VC dimension that

---

[1]In Appendix C.4 we show how to obtain an etimate.

holds for most of these classes. Moreover, we show that for most of these classes the data-generating distribution P is *realizable*. Finally, we design a single active learning algorithm that combines information from all these classes and achieves sublinear learning rates. This algorithm builds upon Hanneke (2012) but is modified to work with partial concept classes instead of total concept classes. This requires a very different analysis and is the most technically involved part of our work.

Let us now explain the main ideas of this algorithm and the challenges behind it. As we mentioned before, the number of queries that the algorithm from Hanneke (2012) needs to achieve the sublinear error rate depends on the underlying data-generating distribution. Thus, we cannot just get a large enough number of unlabeled samples, consider the uniform distribution over them and use the algorithm on this distribution. This is because the learning rate for $\mathbb{H}$ would depend on the uniform distribution $U_S$ over the sample $S$ and not on the data-generating distribution P. To illustrate the ideas of the algorithm, we consider five different streams of i.i.d. (unlabeled) data $S_1, S_2, S_3, S_4, S_5$. For the purposes of the subsequent discussion, we can imagine that these streams have infinite size, but as explained in description of the algorithm, we only need $\mathrm{poly}(n)$ unlabeled points. Let us first explain the use of $S_5$. We use $n/5$ of our query budget to obtain the labels of the first $n/5$ points and then we use them to train a *supervised* learning algorithm from Bousquet et al. (2021) that achieves linear rate $O(1/n)$. This is used for technical reasons in our analysis and in order to ensure that the classifier we output has, at most, linear error rate no matter how the active learning component of our algorithm behaves. Next, we use $n/5$ of the query budget to obtain the labels of the first $n/5$ points from $S_1$. Then, we run the VCL game on these labeled datasets of size $\sqrt{n}/5$ and obtain $\sqrt{n}$ different pattern avoidance functions $\widehat{y}^i_{\sqrt{n}/5}$ that take as input $\ell^i_{\sqrt{n}/5}$ points (cf. Appendix D.2), where $i \in [\sqrt{n}]$. Let

$$\mathcal{F}^i_{\sqrt{n}/5} := \left\{ f : \mathcal{X} \to \{0, 1, \star\} : (f(x_1), \ldots, f(x_{\ell^i_{\sqrt{n}/5}})) \neq \widehat{y}^i_{\sqrt{n}/5}(x_1, \ldots, x_{\ell^i_{\sqrt{n}/5}}) \right\}, i \in [\sqrt{n}].$$

First, we show that for a $(1 - o(1))$-fraction of these partial concept classes P is a realizable distribution. Intuitively, this means that the partial concept class we obtain by running the VCL game on $\sqrt{n}$ many points is the same as the class we would have obtained if we were to run the game on infinitely many points (cf. Lemma D.4). Next, we need to estimate some number $\widehat{d}_n \in \mathbb{N}$ which, as $n \to \infty$, converges to the $9/10$-quantile $d^*$ of the distribution of the VC dimension of the partial concept classes that are obtained by running the VCL game on *infinitely* many samples. We let

$$\widehat{d}_n := \min_{d \in \mathbb{N}} \left\{ \exists i_1, i_2, \ldots, i_{9/10 \cdot \sqrt{n}} \in [\sqrt{n}] : i_1 < i_2 < \ldots < i_{9/10 \cdot \sqrt{n}}, d\left(\mathcal{F}^{i_j}_{\sqrt{n}/5}\right) \leq d, \forall j \in [9/10\sqrt{n}] \right\},$$

where $d(\mathcal{F})$ denotes the VC dimension of class $\mathcal{F}$. Lemma D.5 shows that, for large enough $n$, $\widehat{d}_n = d^*$, with high probability, where $d^* \in \mathbb{N}$ is such that with probability at least $9/10$ over the random draw[2] of the partial class, its VC dimension is at most $d^*$. One technical complication we need to handle is that the concept classes we have obtained are estimated from a game on $\sqrt{n}$ many points instead of infinitely many points, so a $o(1)$-fraction of them do not correspond to samples from the correct distribution. To do that we use a robust version of the well-known Dvoretzky–Kiefer–Wolfowitz (DKW) inequality (cf. Theorem D.1) for estimating the CDF.

Next, we use another $n/5$ of the query budget to obtain the labels of the first $n/5$ points of $S_2$. We will make two distinctions regarding this stream. For the purposes of the analysis, we consider a fixed stream of infinitely many i.i.d. samples from P and we denote it by $S_2^\infty$, but in the actual algorithm we use a dataset of size $\Theta(n)$ and we denote it by $S_2^n$. We define

$$V^i_{n/5} := \left\{ f \in \mathcal{F}^i_{\sqrt{n}/5} : f(x) = y \text{ for the first } n/5 \text{ points in } S_2^\infty \right\},$$

to be the version space of $\mathcal{F}^i_{\sqrt{n}/5}$ defined on the first $n/5$ examples of $S_2^\infty$. Moreover, we let $V_{d^*, n/5}$ be a random version space that is sampled from the following process: we run the VCL game on an infinite stream of labeled data from P to get a pattern avoidance function $\widetilde{y}$ and then we define the partial concept class $\widetilde{\mathcal{F}}$ in the same way as before. If the VC dimension of this class is greater than $d^*$ we discard it and restart the process. Otherwise, we let $V_{d^*, n/5}$ be the version space of $\widetilde{\mathcal{F}}$ on the first $n/5$ labeled points of $S_2^\infty$. We take $d^*$ to be the $9/10-$quantile of $\widetilde{P}$ as described before (cf. Lemma D.5). Given some $k \in \mathbb{N}$, let $p_{n,k} := \mathbb{E}[P^k(x_1, \ldots, x_k \text{ VC shattered by } V_{d^*, n/5}) | S_2^\infty]$ where the expectation is over the draw of $V_{d^*, n/5}$, given the fixed $S_2^\infty$. Let $k^* \in \mathbb{N}$ be the largest number such that

---

[2]This random draw is induced by an execution of the VCL game on infinitely many i.i.d. points from P.

$\lim_{n\to\infty} p_{n,k^*} \neq 0$. Notice that since, by definition, the VC dimension is bounded by $\mathrm{d}^*$ such a number $k^*$ exists. From here on, we will only consider the version spaces $V_{n/5}^i$ that are obtained from some partial class with VC dimension at most $\widehat{\mathrm{d}}_n$. Let $p_{n,k,i} := \mathrm{P}^k(x_1, \ldots, x_k \text{ VC shattered by } V_{n/5}^i)$. Notice that for every $k \leq \widehat{\mathrm{d}}_n$ and every $i \in [\sqrt{n}]$ we can estimate this quantity to arbitrary precision using only *unlabeled* examples. We denote by $\widehat{p}_{n,k,i}$ these estimates.

We now consider the first $n^2$ unlabeled points of the third data stream $S_3$. For each such point $X$, let $p_{n,k}^X := \mathbb{E}[\mathrm{P}^k(x_1, \ldots, x_k, X \text{ VC shattered by } V_{\mathrm{d}^*,n/5}) | S_2^\infty, X]$. Moreover, for each $y \in \{0,1\}$ let

$$V_{\mathrm{d}^*,n/5}^{(X,y)} := \{f \in V_{\mathrm{d}^*,n/5} : f(X) = y\}, \quad p_{n,k}^{(X,y)} := \mathbb{E}[\mathrm{P}^k(x_1, \ldots, x_k \text{ VC shattered by } V_{\mathrm{d}^*,n/5}^{(X,y)}) | S_2^\infty, X], y \in \{0,1\}.$$

We define the quantities $p_{n,k,i}^X, p_{n,k,i}^{(X,y)}$ in the same way for the realized version spaces. Again, by Hoeffding's bound, we can estimate these quantities to arbitrary precision using unlabeled data. Similarly as before, we denote these estimates by $\widehat{p}_{n,k,i}^X, \widehat{p}_{n,k,i}^{(X,y)}$. The idea is to make use of Lemma D.4 and show that the classes which have been obtained by a VCL game that has not converged will only affect our estimates by some $o(1)$. This is formalized in Proposition D.6. Thus, we can use $\widehat{p}_{n,k,i}, \widehat{p}_{n,k,i}^X, \widehat{p}_{n,k,i}^{(X,y)}$, in order to estimate $p_{n,k,}, p_{n,k}^X, p_{n,k}^{(X,y)}$. We denote these estimates by $\widehat{p}_{n,k,}, \widehat{p}_{n,k}^X$, $\widehat{p}_{n,k}^{(X,y)}$. These are the key quantities we use to *infer* the labels of unlabeled points.

Our algorithm tries to infer the label of each unlabeled point $X \in S_3$ (cf. Figure 7) in the following way: if $\widehat{p}_{n,k}^X \geq \frac{\widehat{p}_{n,k}}{2}$ then we query the label of $X$, otherwise we infer the label to be $\arg\max_{y \in \{0,1\}} \widehat{p}_{n,k}^{(X,y)}$. Lemma D.7 shows that the inferences are correct, when $n$ is large enough.

The main ingredient of the proof that remains to be handled is to show that the number of label queries we submit is *sublinear* in $n$. For that, it is sufficient to show that the probability that we query the label of a point is $o(1)$. Lemma D.8 shows that when $k = k^*$, this is indeed the case.

Finally, since we do not know the true value of $k^*$, we run the algorithm for every $k \leq \widehat{\mathrm{d}}_n$. The active learning component of the algorithm gives us $\widehat{\mathrm{d}}_n + 1$ different labeled datasets, which we use to train $\widehat{\mathrm{d}}_n + 1$ instances of a supervised learning algorithm, such as the one from Bousquet et al. (2021). Our analysis so far has shown that for sufficiently large $n$, at least one of these datasets will be correctly labeled with size $\omega(n)$. Thus, since the supervised learning algorithm has error linear in the size of its training set, at least one of these executions will give a classifier that has error $o(1/n)$. The last step is to run a tournament among the $\mathrm{d}_n + 2$ different classifiers[3] to choose the best one. This is handled by Lemma D.9 (Hanneke, 2012). The main result of this section (cf. Theorem D.10), follows as a corollary of the results we have discussed. All the steps are summarized in Figure 7.

Lastly, a lower bound from Hanneke et al. (2022) completes our characterization (cf. Theorem D.11).

## 5 Conclusion

In this work we have provided a complete characterization of the optimal learning rates in active learning. It is an open question if it also holds when the learner knows the distribution $\mathrm{P}_{\mathcal{X}}$.

## Acknowledgments

Amin Karbasi acknowledges funding in direct support of this work from NSF (IIS-1845032), ONR (N00014- 19-1-2406), and the AI Institute for Learning-Enabled Optimization at Scale (TILOS). Shay Moran is a Robert J. Shillman Fellow; he acknowledges support by ISF grant 1225/20, by BSF grant 2018385, by Israel PBC-VATAT, by the Technion Center for Machine Learning and Intelligent Systems (MLIS), and by the the European Union (ERC, GENERALIZATION, 101039692). Views and opinions expressed are however those of the author(s) only and do not necessarily reflect those of the European Union or the European Research Council Executive Agency. Neither the European Union nor the granting authority can be held responsible for them. Grigoris Velegkas was supported in part by the AI Institute for Learning-Enabled Optimization at Scale (TILOS).

---

[3] Recall that we have one more classifier from $S_5$.

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

# A  Omitted Details from Section 1

## A.1  Formal Definition of Learning Setting

We recall some basic notions of measures and probabilities on Polish spaces. For a detailed treatment the reader is referred to Kechris (2012); Cohn (2013). Our presentation follows Bousquet et al. (2021).

**Polish Spaces.**   A *Polish space* is a separable topological space that admits a complete metric. For example, this category includes $\mathbb{R}^n$, any compact metric space, any separable Banach space, etc.

**Universally Measurable Functions.**   Let $\mathfrak{F}$ be the Borel $\sigma$-field on some Polish space $\mathcal{X}$ and let $\mu$ be a probability measure. We denote by $\mathfrak{F}_\mu$ the completion of $\mathfrak{F}$ under $\mu$, i.e., the collections of all subsets of $\mathcal{X}$ that differ from a Borel set on a set of zero measure. A set $B \subseteq \mathcal{X}$ is called *universally measurable* if $B \in \mathfrak{F}_\mu$ for every probability measure $\mu$. Moreover, a function $f : \mathcal{X} \to \mathcal{Y}$ is called universally measurable if $f^{-1}(B)$ is universally measurable, for any universally measurable set $B$. An important property is that universally measurable sets and functions on Polish spaces are the same as Borel sets, from a probabilistic point of view.

We are now ready to provide the definition of measurability of a concept class $\mathbb{H}$.

**Definition A.1** (Measurability of $\mathbb{H}$). *Let $\mathcal{X}$ be a Polish space. We say that a concept class $\mathbb{H} \subseteq \{0,1\}^{\mathcal{X}}$, is measurable if there is a Polish space $\Theta$ and a Borel-measurable map $h : \mathcal{X} \times \Theta \to \{0,1\}$ so that $\mathbb{H} = \{h(\theta, \cdot) : \theta \in \Theta\}$.*

We underline that Definition A.1 is very general and its only requirement is that $\mathbb{H}$ can be parameterized in some reasonable way.

## A.2  Omitted Preliminaries

**Gale-Stewart Games.**   We briefly discuss some basic and useful facts about Gale-Stewart games, a concept that was recently introduced to the learning theory community by Bousquet et al. (2021). Our discussion follows Bousquet et al. (2021); Hanneke et al. (2022). We refer to Bousquet et al. (2021) and references therein for a more detailed presentation. Let us fix sequences of sets $\mathcal{X}_t, \mathcal{Y}_t$ for $t \geq 1$. We consider infinite games between two players, a learner $P_L$ and an adversary $P_A$, where in each round $t \geq 1$, $P_A$ selects an element $x_t \in \mathcal{X}_t$, and then $P_L$ selects an element $y_t \in \mathcal{Y}_t$. The rules of the game are defined by a set $\mathcal{W} \subseteq \prod_{t \geq 1}(\mathcal{X}_t \times \mathcal{Y}_t)$ of winning sequences for $P_L$. This means that after an infinite sequence of consecutive plays $x_1, y_1, x_2, y_2, \ldots$, we say that $P_L$ wins if $(x_1, y_1, x_2, y_2, \ldots) \in \mathcal{W}$; otherwise the winner is $P_A$.

A *strategy* is a rule used by each of the players to determine their next move, given the current state and the history of the game. Formally, a strategy for $P_A$ is a sequence of functions $f_t : \prod_{s<t}(\mathcal{X}_s \times \mathcal{Y}_s) \to \mathcal{X}_t$ for $t \geq 1$, so that $P_A$ plays $x_t = f_t(x_1, y_1, \ldots, x_{t-1}, y_{t-1})$ in round $t$. Similarly, a strategy for $P_L$ is a sequence of $g_t : \prod_{s<t}(\mathcal{X}_s \times \mathcal{Y}_s) \times \mathcal{X}_t \to \mathcal{Y}_t$ for $t \geq 1$, so that $P_L$ plays $y_t = g_t(x_1, y_1, \ldots, x_{t-1}, y_{t-1}, x_t)$ in round $t$. We say that a strategy for $P_A$ is *winning* if playing that strategy always makes $P_A$ win the game no matter what $P_L$ plays; a winning strategy for $P_L$ is defined similarly. The main question in these infinite games is determine conditions under which one of the two players has a winning strategy in the game. We remark that when the game is finite it is not hard to easy that exactly one of the two players has such a strategy. In the context of infinite games, such a condition was introduced by Gale and Stewart (1953): a $\mathcal{W}$ is *finitely decidable* if for every sequence of plays $(x_1, y_1, x_2, y_2, \ldots) \in \mathcal{W}$, there exists some $n < \infty$ so that

$$(x_1, y_1, \ldots, x_n, y_n, x'_{n+1}, y'_{n+1}, x'_{n+2}, y'_{n+2}, \ldots) \in \mathcal{W}$$

for all choices of $x'_{n+1}, y'_{n+1}, x'_{n+2}, y'_{n+2}, \ldots$ In words, the condition that "$\mathcal{W}$ is finitely decidable" means that if $P_L$ wins the game, then she knows that after playing a *finite* number of rounds. Conversely, $P_A$ wins the game when $P_L$ does not win after any finite number of rounds.

Such a game whose set $\mathcal{W}$ is finitely decidable is called a *Gale-Stewart game*. We use the following important result about Gale-Stewart games.

**Remark A.2** (Gale and Stewart (1953); Hodges et al. (1993); Kechris (2012)). *In any Gale-Stewart game exactly one between $P_A$ and $P_L$ has a winning strategy.*

The above result is purely existential and provides no information about the complexity of the winning strategies. Importantly, it is unclear whether winning strategies can be chosen to be measurable. The next result from Bousquet et al. (2021) addresses this concern.

**Theorem A.3** (Theorem B.1 from Bousquet et al. (2021)). *Let $\{X_t\}_{t\geq 1}$ be Polish spaces and $\{Y_t\}_{t\geq 1}$ be countable sets. Consider a Gale-Stewart game whose set $\mathcal{W} \subseteq \prod_{t\geq 1}(X_t \times Y_t)$ of winning strategies for $P_L$ is finitely decidable and coanalytic. Then there is a universally measurable winning strategy.*

The following remark shows that we can, equivalently, let the strategy of the the learner $P_L$ and the adversary $P_A$ depend only on the choices of their opponent in the previous rounds.

**Remark A.4** (Bousquet et al. (2021)). *The strategy of $P_L$ is defined to be a sequence of functions $y_t = f_t(x_1, y_1, \ldots, x_{t-1}, y_{t-1})$ of the history of the game, where $y_1, \ldots, y_{t-1}$ are defined similarly. Thus, we can equivalently let $y_t = f_t(x_1, \ldots, x_{t-1})$. The same holds for the strategy of $P_A$.*

**Partial Concept Classes.** The traditional PAC learning framework studies (mainly) *total* concept classes, i.e., classes of functions $h : \mathcal{X} \to \{0, 1\}$ that are defined on *every* point $x \in \mathcal{X}$. The caveat with total functions is that they do not provide a direct way to express *data-dependent* constraints. For example, if the space $\mathcal{X}$ is high-dimensional but the data that the leaner has to classify lie in a low-dimensional space, it is not clear how to encode this restriction through total concepts. This is very relevant to practical application of machine learning such as classification of images; the space $\mathcal{X}$ is the set of all possible values of the pixels of the image but most of these configurations of the pixels do not even correspond to an image of an object of interest. Alon et al. (2021) proposed and studied an extension of the PAC framework that allows one to capture such assumptions using *partial* concept classes, i.e., sets of functions $f : \mathcal{X} \to \{0, 1, \star\}$, where $f(x) = \star$ means that $f$ is undefined at $x$. In the context of classification of images, when a classifier returns $\star$ it means that $x$ does not belong to the space of valid images. Alon et al. (2021) extend a lot of notions, such as PAC learnability and the VC dimension, from the setting of total concept classes to the setting of partial concept classes (see, e.g., Definition A.6). To show how one can use partial classes, to express data-dependent constraints, we remark that Alon et al. (2021) illustrated how the class of $d$-dimensional linear classifiers with margin $\gamma > 0$ can be formulated as a partial class: we say that a sample $(x_1, y_1), \ldots, (x_n, y_n) \in \mathbb{R}^d \times \{0, 1\}$ is $(R, \gamma)$-separable if all the points $x_1, \ldots, x_n$ lie in a (euclidean) ball of radius $R$, the 0-labeled examples and the 1-labeled examples are linearly separable, and the (euclidean) distance between the 0-labeled examples and 1-labeled examples is at least $2\gamma$. Then, the class

$$\mathcal{F}_{R,\gamma} = \big\{ f : \mathbb{R}^d \to \{0, 1\star\} : (\forall x_1, \ldots, x_n) \in \mathrm{supp}(f) :$$
$$(x_1, (f(x_1)), \ldots, (x_n, (f(x_n)) \text{ is } (R, \gamma) - \text{separable} \big\},$$

where $\mathrm{supp}(f)$ is the set of all points where $f(x) \neq \star$, expresses the set of functions that satisfy these constraints. Remarkably, the VC dimension of $\mathcal{F}$ is bounded by $O\left(\frac{R^2}{\gamma^2}\right)$ (Alon et al., 2021). Perhaps surprisingly, even though the PAC learnability of partial classes is characterized by the VC dimension (as it is the case with total classes), the ERM algorithm provably fails to learn partial classes. Thus, the algorithmic landscape is much richer and complicated compared to total classes. Moreover, there is no algorithmic way to extend a partial concept class to a total concept class without significantly increasing its VC dimension. For details, we refer to (Alon et al., 2021).

**One-Inclusion Graph Algorithm.** We state formally the guarantees of the one-inclusion graph algorithm (Haussler et al., 1994).

**Theorem A.5** (One-Inclusion Graph Algorithm (Haussler et al., 1994)). *For any (total) concept class $\mathbb{H}$ whose VC dimension is bounded by $\mathrm{d} < \infty$, there is an algorithm $\mathbb{A} : (\mathcal{X} \times \{0, 1\})^* \times \mathcal{X} \to \{0, 1\}$ such that for any $n \in \mathbb{N}$ and any sequence $\{(x_1, y_1), \ldots, (x_n, y_n)\} \in (\mathcal{X} \times \{0, 1\})^n$ that is realizable w.r.t. $\mathbb{H}$,*

$$\frac{1}{n!} \sum_{\sigma \in \mathrm{Sym}(n)} \mathbb{1}\{\mathbb{A}(x_{\sigma(1)}, y_{\sigma(1)}, \ldots, x_{\sigma(n-1)}, y_{\sigma(n-1)}, x_{\sigma(n)}) \neq y_{\sigma(n)}\} \leq \frac{\mathrm{d}}{n},$$

*where $\mathrm{Sym}(n)$ denotes the symmetric group of permutations of $\{1, \ldots, n\}$.*

In particular, Theorem A.5 implies immediately that if $(x_1, y_1), \ldots, (x_n, y_n)$ are i.i.d. from P then the classifier $\tilde{h}_n(\cdot) := \mathbb{A}(x_1, y_1, \ldots, x_n, y_n, \cdot)$ has $\mathbb{E}[\mathrm{er}(\tilde{h}_n)] \leq \frac{\mathrm{d}}{n+1}$. We remark that Alon et al. (2021) showed that this result also holds for partial concept classes.

## A.3 Omitted Definitions

**Definition A.6** (VC Dimension of Partial Concept Classes (Alon et al., 2021)). *For a partial concept class $\mathcal{F} \subseteq \{0, 1, \star\}^{\mathcal{X}}$, the VC dimension of $\mathcal{F}$ is defined to be the largest number $\mathrm{d} \in \mathbb{N}$ such that $\exists (x_1, \ldots, x_{\mathrm{d}}) \in \mathcal{X}^{\mathrm{d}}$ such that $\{(f(x_1), \ldots, f(x_{\mathrm{d}})) : f \in \mathcal{F}\} = \{0, 1\}^{\mathrm{d}}$. Such a sequence $(x_1, \ldots, x_{\mathrm{d}})$ is said to be shattered by $\mathcal{F}$. If there is no bound on $\mathrm{d}$ we say that the VC dimension is $\infty$.*

The following definition of the star number is an adaptation of the definition in (Hanneke and Yang, 2015).

**Definition A.7** (Star Number of Partial Concept Classes). *For a partial concept class $\mathcal{F} \subseteq \{0, 1, \star\}^{\mathcal{X}}$, the star number of $\mathcal{F}$ is defined to be the largest number $\mathfrak{s} \in \mathbb{N}$ such that $\exists (x_1, \ldots, x_{\mathfrak{s}}) \in \mathcal{X}^{\mathfrak{s}}$ such that $\exists f_0 \in \mathcal{F}$ and $\forall i \in [\mathfrak{s}], \exists f_i \in \mathcal{F} : f_i(x_i) = 1 - f_0(x_i), f_i(x_j) = f_0(x_j) \neq \star, \forall j \in [\mathfrak{s}] \setminus \{i\}$. If there is no bound on $\mathfrak{s}$ we say that the star number is $\infty$.*

**Definition A.8** (Littlestone Tree (Bousquet et al., 2021)). *A Littlestone tree for $\mathbb{H} \subseteq \{0, 1\}^{\mathcal{X}}$ is a complete binary tree of depth $d \leq \infty$ whose internal nodes are labeled by $\mathcal{X}$, and whose two edges connecting a node to its children are labeled by $\{0, 1\}$, such that every path of length at most $d$ emanating from the root is consistent with a concept $h \in \mathbb{H}$. Formally, a Littlestone tree is a collection*

$$\bigcup_{0 \leq \ell < d} \{x_{\vec{u}} : \vec{u} \in \{0, 1\}^{\ell}\} = \{x_{\emptyset}\} \cup \{x_0, x_1\} \cup \{x_{00}, x_{01}, x_{10}, x_{11}\} \cup \ldots$$

*such that for every path $\vec{y} \in \{0, 1\}^d$ and $n < d$, there exists $h \in \mathbb{H}$ so that $h(x_{\vec{y}_{\leq \ell}}) = y_{\ell+1}$ for $0 \leq \ell \leq n$. We say that $\mathbb{H}$ has an infinite Littlestone tree if there is a Littlestone tree for $\mathbb{H}$ of depth $d = \infty$.*

**Definition A.9** (Star Tree). *A Star tree for $\mathbb{H} \subseteq \{0, 1\}^{\mathcal{X}}$ of consists of a collection*

$$\bigcup_{0 \leq \ell < d} \left\{ (x_{\vec{u}}, z_{\vec{u}}) \in \left( \mathcal{X}^{\ell+1}, \{0, 1\}^{\ell+1} \right) : \vec{u} \in \{0\} \times \{0, 1\} \times \ldots \times \{0, 1, \ldots, \ell\} \right\}$$

*such that for every level $n < d$, and $\vec{y} \in \{0\} \times \{0, 1\} \times \ldots \times \{0, 1, \ldots, n-1\}$, there exists some $h \in \mathbb{H}$ such that $h(x^i_{\vec{y}_{\leq k}}) = f(z^i_{\vec{y}_{\leq k}}, y_{k+1})$ for all $0 \leq i \leq k$ and $0 \leq k \leq n$, where we denote*

$$\vec{y}_{\leq k} = (y_1, y_2, \ldots, y_k), \quad x_{\vec{y}_{\leq k}} = (x^0_{\vec{y}_{\leq k}}, \ldots, x^k_{\vec{y}_{\leq k}}), \quad z_{\vec{y}_{\leq k}} = (z^0_{\vec{y}_{\leq k}}, \ldots, z^k_{\vec{y}_{\leq k}}),$$

$$f(z^i_{\vec{y}_{\leq k}}, y_{k+1}) = \begin{cases} z^i_{\vec{y}_{\leq k}}, & \text{if } y_{k+1} \neq i \\ 1 - z^i_{\vec{y}_{\leq k}}, & \text{otherwise} \end{cases}.$$

*We say that $\mathbb{H}$ has an infinite star tree if it has a star tree of depth $d = \infty$.*

**Definition A.10** (VCL Tree Bousquet et al. (2021)). *A Vapnik-Chervonenkis-Littlestone (VCL) tree for $\mathbb{H} \subseteq \{0, 1\}^{\mathcal{X}}$ of depth $d \leq \infty$ consists of a collection*

$$\bigcup_{0 \leq \ell < d} \{x_{\vec{u}} \in \mathcal{X}^{\ell+1}, \vec{u} \in \{0, 1\} \times \{0, 1\}^2 \times \ldots \times \{0, 1\}^{\ell}\}$$

*such that for every level $n < d$, and $\vec{y} \in \{0, 1\} \times \{0, 1\}^2 \times \{0, 1\}^{n+1}$, there exists some $h \in \mathbb{H}$ such that $h(x^i_{\vec{y}_{\leq k}}) = y^i_{k+1}$ for all $0 \leq i \leq k$ and $0 \leq k \leq n$, where we denote*

$$\vec{y}_{\leq k} = (y^0_1, (y^0_2, y^1_2), \ldots, (y^0_k, \ldots, y^{k-1}_k)), \quad x_{\vec{y}_{\leq k}} = (x^0_{\vec{y}_{\leq k}}, \ldots, x^k_{\vec{y}_{\leq k}}).$$

*We say that $\mathbb{H}$ has an infinite VCL tree if it has a VCL tree of depth $d = \infty$.*

Figure 1: Arbitrarily Fast Rates Algorithm (Hanneke et al., 2022)

.

---

**Arbitrarily Fast Rates (Hanneke et al., 2022):** Input is a label budget $N$ and a rate function $R(n)$.

1. $m_1 \leftarrow \lceil 2\ln(1/R(n)) \rceil$.

2. $m_2 \leftarrow \left\lceil \frac{\ln\left(m_1 2^{2\lfloor\sqrt{m_1}\rfloor+2}/R(n)\right)}{R(n)} \right\rceil$.

3. Create sets $S_1 = x_1, \ldots, x_{m_1}$, $S_2 = x_{m_1+1}, \ldots, x_{m_2}$.

4. Split $S_1$ into $\lfloor\sqrt{m_1}\rfloor$ batches of size $\lfloor\sqrt{m_1}\rfloor$ and consider all the labeled prefixes of the batches in lexicographical order: $b_{i,j}$ denotes the $i$-th batch with the labeled prefix $j$, $1 \leq j \leq 2^{\lfloor\sqrt{m_1}\rfloor+1}$.

5. Let $\mathbb{A}_0(\cdot; b_{i,j})$ be the output of the ordinal SOA trained on $b_{i,j}$ and denote $g_{i,j}(x) = \mathbb{A}_0(x; b_{i,j})$.                See (Bousquet et al., 2021).

6. Evaluate the classifiers $g_{i,j}(\cdot)$ on the points in $S_2$.

7. Define a set $\mathbb{F}$ of equivalence classes: $g_{i,j}$ and $g_{i',j'}$ are in the same class iff they classify $S_2$ the same.

8. For each $F \in \mathbb{F}$, define

$$\mathrm{rank}(F) = \min\left\{ r : \exists\, 1 \leq i_1 < i_2 \ldots < i_{\lfloor\sqrt{m_1}\rfloor/3} \leq \lfloor\sqrt{m_1}\rfloor, k_1, \ldots, k_{\lfloor\sqrt{m_1}\rfloor/3} \in [2^{r+1}] \right.$$
$$\left. \text{such that } g_{i_j,k_j} \in F, 1 \leq j \leq \lfloor\sqrt{m_1}\rfloor/3 \right\},$$

or $\mathrm{rank}(F) = \infty$ if no such $r$ exists.

9. Enumerate $\mathbb{F} = \{\mathcal{F}_1, \mathcal{F}_2, \ldots\}$ so that $\mathrm{rank}(\mathcal{F}_\ell)$ is non-decreasing in $\ell$, and pick any $f_\ell \in \mathcal{F}_\ell$ for each $\ell$.

10. For each distinct $i, j \leq \lfloor\sqrt{n}\rfloor$ query the label of any point in $S_2$ on which $f_i, f_j$ disagree.

11. If there is some $f_{\hat\ell}$ which is correct on all the points we queried, return $f_{\hat\ell}$.

12. Otherwise, return any $f_\ell$.

---

# B  Omitted Details from Section 2

# C  Omitted Details from Section 3

## C.1  Proof of Lemma 3.1

Before we prove our main result of this section, we present an omitted proof from the main body.

*Proof.* Assume that $\mathbb{H}$ has an infinite star tree. Then, the adversary can start from the root of the tree and play $(\vec{\xi}_1, \vec{\zeta}_1) = (\vec{x}_\varnothing, \vec{z}_\varnothing)$ and for every strategy $\{\eta_\tau\}_{\tau \geq 1}$ of the learner, $P_A$ can follow the path $\vec{y}_{\leq\tau} = (\eta_1, \eta_2, \ldots, \eta_\tau)$ and play $(\vec{x}_{\vec{y}_{\leq\tau}}, \vec{z}_{\vec{y}_{\leq\tau}})$ in round $\tau + 1$. Since the tree is infinite, there is no $\tau^* \in \mathbb{N}$ such that $\mathbb{H}_{\vec{\xi}_1, \vec{\zeta}_1\eta_1, \ldots, \vec{\xi}_{\tau^*}, \vec{\zeta}_{\tau^*}, \eta_{\tau^*}} = \varnothing$.

For the other direction, assume that the adversary has a winning strategy. Then, we can define the tree

$$\mathcal{T} = \bigcup_{0 \leq \ell < \infty} \left\{ (\vec{x}_{\vec{\eta}}, \vec{z}_{\vec{\eta}}) \in \left(\mathcal{X}^{\ell+1}, \{0,1\}^{\ell+1}\right) : \vec{\eta}_i \in \{0, 1, \ldots, i-1\}, 1 \leq i \leq \ell \right\}$$

where $(\vec{x}_{\eta_1, \ldots, \eta_{\tau-1}}, \vec{z}_{\eta_1, \ldots, \eta_{\tau-1}}) = (\vec{\xi}_\tau(\eta_1, \ldots, \eta_{\tau-1}), \vec{\zeta}_\tau(\eta_1, \ldots, \eta_{\tau-1}))$. The tree $\mathcal{T}$ is infinite by the definition of the winning strategy in $\mathfrak{G}$ for $P_A$. To prove the universal measurability of the winning strategy of $P_L$, using Lemma A.3 it suffices to show that the set of winning strategies $\mathcal{W}$ of the learner

is coanalytic. To this end, we consider its complement

$$\mathcal{W}^c = \left\{ (\vec{\xi}, \vec{\zeta}, \vec{\eta}) \in \bigcup_{t=1}^{\infty} \left( \mathcal{X}^t \times \{0,1\}^t \times \{0,\ldots,t-1\} \right) : \mathbb{H}_{\vec{\xi}_1, \vec{\zeta}_1, \eta_1, \ldots, \vec{\xi}_t, \vec{\zeta}_t, \eta_t} \neq \varnothing \right\}$$

$$= \bigcap_{t=1}^{\infty} \bigcup_{\theta \in \Theta} \bigcap_{s=1}^{t} \left\{ (\vec{\xi}, \vec{\zeta}, \vec{\eta}) : \begin{array}{ll} h(\theta, \vec{\xi}_s^i) = \vec{\zeta}_s^i, & \text{if } \eta_s \neq i \\ h(\theta, \vec{\xi}_s^i) = 1 - \vec{\zeta}_s^i, & \text{if } \eta_s = i \end{array}, 0 \leq i < s \right\}$$

Notice that the set $\left\{ \begin{array}{ll} h(\theta, \vec{\xi}_s^i) = \vec{\zeta}_s^i, & \text{if } \eta_s \neq i \\ h(\theta, \vec{\xi}_s^i) = 1 - \vec{\zeta}_s^i, & \text{if } \eta_s = i \end{array}, 0 \leq i < s \right\}$ is Borel by the measurability assumption. Also, the intersection in the above expression are countable and the union is a projection of a Borel set. Hence, $\mathcal{W}^c$ is analytic, so we have proven the claim. □

## C.2 Active Learning of Partial Classes with Finite Star Number: Exponential Uniform Rate

We now explain the approach to actively learn a partial concept class with finite star number at a *uniform* exponential rate. We first state a result from Hanneke and Yang (2015) about active learning of *total* concept classes with finite star dimension that we utilize in our construction.

**Theorem C.1** (Hanneke and Yang (2015)). *There exists an active learning algorithm $\mathcal{A}$ for a total concept class $\mathbb{H}$ which given a label budget $n = c \cdot \mathfrak{s} \log(1/\varepsilon)$, where $c$ is an absolute numerical constant and $\mathfrak{s}$ is the star number of the class, and access to samples from a realizable distribution $\mathrm{P}^*$ returns a classifier $\hat{h}_n$ such that, with probability arbitrarily close to one, $\mathrm{P}^*[\hat{h}_n(x) \neq y] \leq \varepsilon$.*

We underline that the success probability of this algorithm does not depend on the query budget.

Our algorithm for partial classes can be found in Figure 2. We now prove the main result in this setting that we described in Theorem 3.2.

*Proof of Theorem 3.2.* Let $\mathcal{F}$ be a partial concept class whose star number $\mathfrak{s}$ is finite. Notice that since $\mathrm{d} \leq \mathfrak{s}$, its VC dimension is also finite. Let $S = (X_1, \ldots, X_m)$ be $m$ samples of the stream that $\mathcal{A}$ has access to, where $m$ is some value to be determined. We also let

$$\widehat{\mathbb{H}} = \{h : S \to \{0,1\} : \exists f \in \mathcal{F} \text{ s.t. } h(X_i) = f(X_i), \forall i \in [m]\} .$$

Now notice that $\widehat{\mathbb{H}}$ is, by definition, a total concept class and its star number, VC dimension is also at most $\mathfrak{s}, d$, respectively. Moreover, since $\mathrm{P}^*$ is realizable with respect to $\mathcal{F}$ we have that, except for a measure-zero event, $\widehat{\mathbb{H}} \neq \emptyset$. This can be seen as follows. Since the distribution $\mathrm{P}^*$ is realizable, there exists a sequence of functions $f_k \in \mathcal{F}$ so that

$$\mathrm{P}^*[f_k(x) \neq y] < \frac{1}{2^k} .$$

Let us fix $m \geq 1$. We have that

$$\sum_{k=1}^{\infty} \Pr[\exists s \leq m : f_k(x_s) \neq y_s] \leq m \sum_{k=1}^{\infty} \mathrm{P}^*[f_k(x) \neq y] < \infty ,$$

where the first inequality is due to union bound. By Borel-Cantelli, with probability one, there exists for every $m \geq 1$ a hypothesis $f \in \mathcal{F}$ so that $f(x_s) = y_s$ for all $s \leq m$. For the rest of the proof, we condition on this probability one event $E_0$. We definite $\widehat{\mathrm{P}}^*$ to be the uniform distribution on the (labeled) sample. Our previous discussion shows that $\widehat{\mathrm{P}}^*$ is realizable with respect to $\widehat{\mathbb{H}}$. Now let us run algorithm $\mathcal{A}'$ from Theorem C.1 with $\varepsilon = 1/(2m)$ and success probability $1/(2m)$. Let $E_1$ be the event that the error of the output $\tilde{h}$ of $\mathcal{A}'$ is at most $1/(2m)$. We condition on this event for the rest of the proof. Notice that in this case, the algorithm will have figured out the correct label of every $X_i, 1 \leq i \leq m$. Indeed, if for some $i \in [m]$ we have that $\tilde{h}(X_i) \neq Y_i$, then $\mathrm{er}(\tilde{h}) \geq 1/m$. Notice that, by the definition of $\widehat{\mathbb{H}}$, we have that $\mathfrak{s}(\widehat{\mathbb{H}}) \leq \mathfrak{s}(\mathcal{F})$. Moreover, the number of labels that $\mathcal{A}'$ requests is at most $c \cdot \mathfrak{s} \log(2m)$, where $c$ is some absolute numerical constant.

Since we have $m$ *labeled* samples, we can use them as input to the one-inclusion graph algorithm (see Theorem A.5). This algorithm guarantees that

$$\mathbb{E}[\mathrm{er}(\hat{h}_m)|E_0, E_1] \leq \frac{\mathrm{d}(\mathcal{F})}{m+1} .$$

Thus, we get that

$$\mathbb{E}[\mathrm{er}(\hat{h}_m)] \leq \frac{\mathrm{d}(\mathcal{F})}{m+1} + \frac{1}{m} < \frac{\mathrm{d}(\mathcal{F})+1}{m} \,.$$

It follows immediately from the definitions that $\mathrm{d}(\mathcal{F}) \leq \mathfrak{s}(\mathcal{F})$. The result follows by picking $m = \frac{e^{c \cdot n/\mathfrak{s}}}{2}$. $\qquad \square$

---

**Exponential Rates for Partial Classes**

1. $m \leftarrow \frac{e^{c \cdot n/\mathfrak{s}}}{2}$, where $c$ is the constant in Theorem C.1.
2. Take a sample of unlabeled data $S = \{X_1, \ldots, X_m\}$.
3. Define $\widehat{\mathbb{H}}$ to be the total class of the concepts in $\mathcal{F}$ that are *not* undefined on $S$, i.e.,

$$\widehat{\mathbb{H}} = \{h : S \to \{0,1\} : \exists f \in \mathcal{F} \text{ s.t. } h(X_i) = f(X_i), \forall i \in [m]\} \,.$$

4. Let $\widehat{\mathrm{P}}$ be the uniform distribution over the sample $\{(X_1, Y_1), \ldots, (X_m, Y_m)\}$, where $Y_i$ is the true label of $X_i, i \in [m]$.
5. Use the active learning algorithm from Hanneke and Yang (2015) on the distribution $\widehat{\mathrm{P}}$ (see Theorem C.1).
6. The aforementioned algorithm can find the labels of these $m$ points after making at most $O(\mathfrak{s}(\widehat{\mathbb{H}}) \log m)$ queries, where $\mathfrak{s}(\widehat{\mathbb{H}})$ is the star number of $\widehat{\mathbb{H}}$.
7. Use these $m$ labeled points as input to the one-inclusion graph algorithm (Haussler et al., 1994) and get $\hat{h}$ (see Theorem A.5).
8. This algorithm guarantees linear error in $m$, i.e., $\mathbb{E}[\mathrm{er}(\hat{h})] \leq \frac{\mathrm{d}(\mathcal{F})}{m+1}$, where $\mathrm{d}(\mathcal{F})$ is the VC dimension of $\mathcal{F}$.
9. Return $\hat{h}_m$.

Figure 2: Exponential Rates Algorithm for Partial Classes.

---

**Star Tree Gale-Stewart Game with Finite Sequence and Estimation:** Input is a labeled sequence $\{(x_1, y_1), \ldots, (x_N, y_N)\}$

1. Use $\lfloor N/2 \rfloor$ of the data to estimate $\hat{t}_N$ (cf. Lemma C.5.). Let $\hat{N} = \lfloor N/2\hat{t}_N \rfloor$.
2. Use the remaining $\lfloor N/2 \rfloor$ of the data to run the star tree game (cf. Figure 5) and obtain $\tilde{g}^i_{\hat{t}_N}, 1 \leq i \leq \hat{N}$.
3. Estimate for $1 \leq i \leq \hat{N}$

$$\mathcal{F}_i = \left\{ f : \mathcal{X} \to \{0,1,*\} : \forall (x_1, \ldots, x_{\tau_{\hat{t}_N}}) \in \mathcal{X}^{\tau_{\hat{t}_N}}, \left( x_1, f(x_1), \ldots, x_{\tau_{\hat{t}_N}}, f(x_{\tau_{\hat{t}_N}}) \right) \notin \mathrm{image}\left( \tilde{g}^i_{\hat{t}_N} \right) \right\}.$$

4. Estimate the $9/16-$majority class

$$\mathcal{F}^{\hat{N}}_m = \left\{ f : \mathcal{X} \to \{0,1,\star\} : \forall \ell \in \mathbb{N}, \forall (x_1, \ldots, x_\ell) \in \mathcal{X}^\ell, \exists \text{ at least } (9/16)\hat{N} \text{ classes } \mathcal{F}_j, 1 \leq j \leq \hat{N} \text{ s.t.} \right.$$

$$\left. \exists \hat{f} \in \mathcal{F}_j \text{ with } (\hat{f}(x_1), \ldots, \hat{f}(x_\ell)) = (f(x_1), \ldots, f(x_\ell)) \right\}.$$

5. Return $\left\{ G_\ell : (\mathcal{X} \times \{0,1\})^\ell \to \{0,1\} \right\}_{\ell \in \mathbb{N}}$ where
$$G_\ell(x_1, y_1, \ldots, x_\ell, y_\ell) = \mathbb{1}\left\{ \exists f \in \mathcal{F}^{\hat{N}}_m : (f(x_1), \ldots, f(x_\ell)) = (y_1, \ldots, y_\ell) \right\}.$$

Figure 3: Majority Class Estimation Through Star Games.

Figure 4: Exponential Rates Algorithm.

---

**Exponential Rates**: Input is an unlabeled data stream $\{x_1, x_2, \ldots, \}$ and query budget $n$.

1. Set number of unlabeled examples $u_n = \lfloor n/2 \rfloor + e^{c \cdot n}$, where $c$ is the absolute constant in Theorem C.1.

2. Use $\lfloor n/2 \rfloor$ label queries to get the labels of the first $\lfloor n/2 \rfloor$ unlabeled points.

3. Call the Gale-Stewart subroutine (Figure 3) with points
$$\left\{ (x_1, y_1), \ldots, \left( x_{\lfloor n/2 \rfloor}, y_{\lfloor n/2 \rfloor} \right) \right\}.$$

4. Create the partial concept class
$$\mathcal{F}_G = \left\{ f : \mathcal{X} \to \{0, 1, \star\} : \forall \ell \in \mathbb{N}, \forall (x_1, \ldots, x_\ell) \in \mathcal{X}^\ell, G_\ell(x_1, f(x_1), \ldots, x_\ell, f(x_\ell)) = 1 \right\}.$$

5. Run Figure 2 on the partial class $\mathcal{F}_G$ with budget $\lfloor n/2 \rfloor$ and return its output.

---

### C.3 The Online Setting

To build some intuition, we first show how to use the game between the learner and the adversary discussed in Section 3 to get an asymptotic result. We call this the online setting. Assume that there is an infinite labeled sequence $S = (x_1, y_1, x_2, y_2, \ldots)$ that is consistent with $\mathbb{H}$, i.e., for any $t < \infty$, there exists $h \in \mathbb{H} : h(x_t) = y_t$. We use $S$ to simulate the game $\mathfrak{G}$ between $P_A$ and $P_L$ which gives rise to a partial concept class $\mathcal{F}$ with finite star number. This approach is shown in Figure 5.

---

**Star Gale-Stewart Game with Infinite Sequence**

1. Let $g_1 : \mathcal{X} \times \{0, 1\} \to \{0, 1\}$ be the function that corresponds to the strategy of $P_L$ in the first round of $\mathfrak{G}$.

2. Initialize $\tau_0 \leftarrow 1$.

3. Let $\widetilde{g}_1(x, y) = (x, 1 - y)$        # In the first round the learner just flips the label.

4. For every $t \geq 1$:

   • If $(x_{t-\tau_{t-1}+1}, y_{t-\tau_{t-1}+1}, \ldots, x_t, y_t) \in \text{image}(\widetilde{g}_{\tau_{t-1}})$, witnessed by $\vec{c}'_{\tau_{t-1}}$:
   # proceed to the next round in $\mathfrak{G}$.

     – $\tau_t \leftarrow \tau_{t-1} + 1$.

     – $g_{\tau_t}(\cdot, \cdot) := \eta_{\tau_t}(\vec{c}'_1, \ldots, \vec{c}'_{\tau_{t-1}} \cdot, \cdot)$, where $\eta_{\tau_t}(\vec{c}'_1, \ldots, \vec{c}'_{\tau_{t-1}} \cdot, \cdot)$ is the strategy of $P_L$ in round $\tau_t$ of $\mathfrak{G}$, given its history.

     – $\widetilde{g}_{\tau_t}(x_0, y_0, \ldots, x_{\tau_t - 1}, y_{\tau_t - 1}) := (x_0, \widetilde{y}_0, \ldots, x_{\tau_t - 1}, \widetilde{y}_{\tau_t - 1}) :$
     $$\left\{ \begin{array}{ll} \widetilde{y}_i = y_i, & \text{if } g_{\tau_t}(x_0, y_0, \ldots, x_{\tau_t - 1}, y_{\tau_t - 1}) \neq i \\ \widetilde{y}_i = 1 - y_i, & \text{if } g_{\tau_t}(x_0, y_0, \ldots, x_{\tau_t - 1}, y_{\tau_t - 1}) = i \end{array} \right., 0 \leq i < \tau_t \right\}$$

   • Else: $\tau_t \leftarrow \tau_{t-1}$.

---

Figure 5: Star Gale-Stewart Game on Infinite Sequences.

We first show as $t \to \infty$ the game converges, i.e., there are no more realizable $\tau_{t^*}$-tuples $(x_1, y_1, \ldots, x_{\tau_{t^*}}, y_{\tau_{t^*}})$ that belong to the image of $\widetilde{g}_{\tau_{t^*}}$, for some finite number $t^*$. To simplify the notation, we define $\widetilde{g}_{t^*} = \widetilde{g}_{\tau_{t^*}}$. Recall that this function is defined in Figure 5.

**Lemma C.2.** *If $\mathbb{H}$ does not have an infinite star tree, for any sequence $x_1, y_1, x_2, y_2, \ldots$, that is consistent with $\mathbb{H}$, there exists some finite number $t^* \in \mathbb{N}$, such that for all $t > t^*$*

$$(x_{t-\tau_{t-1}+1}, y_{t-\tau_{t-1}+1} \ldots, x_t, \ldots, y_t) \notin \text{image}(\widetilde{g}_{t-1}), \tau_t = \tau_{t-1}, \widetilde{g}_t = \widetilde{g}_{t-1}.$$

*Proof.* Suppose that $(x_{t-\tau_{t-1}+1}, \ldots, x_t, y_{t-\tau_{t-1}+1}, \ldots, y_t) \in \text{image}(\widetilde{g}_{t-1})$ happens an infinite sequence of times $t_1, t_2, \ldots$. Since $\eta_{\tau_t}$ is a winning strategy for the learner in $\mathfrak{G}$, we have that for some finite $t^*$ it holds that $\mathbb{H}_{\vec{\xi}_1, \vec{\zeta}_1, \eta_1, \ldots, \vec{\xi}_{t^*}, \vec{\zeta}_{t^*}, \eta_{t^*}} = \varnothing$ (see Equation (1)), where $\vec{\xi}_i, \vec{\zeta}_i, \eta_i$ are defined in Figure 5. Hence, we have arrived in a contradiction. $\square$

Similarly as in (Bousquet et al., 2021), we remark the following about the measurability of the functions.

**Remark C.3** (Adapted from Bousquet et al. (2021)). *The strategy $\tau_t$ depends in a universally measurable way on $x_{\leq t}, y_{\leq t}$. Moreover, the functions $\widetilde{g}_t$ are universally measurable with respect to $x_{\leq t}, y_{\leq t}$ and the arguments of their input. To be more precise, for all $t \geq 0$, there exist the following universally measurable functions*

$$T_t : (\mathcal{X} \times \{0,1\})^t \to \{1, \ldots, t+1\}, \quad \widetilde{G}_t : (\mathcal{X} \times \{0,1\})^t \times \left( \bigcup_{s<t} (\mathcal{X} \times \{0,1\})^s \right) \to (\mathcal{X} \times \{0,1\})^t ,$$

*so that*

$$\tau_t = T_t(x_1, y_1, \ldots, x_t, y_t), \quad \widetilde{g}_t(w_1, z_1, \ldots, w_{\tau_t}, z_{\tau_t}) = \widetilde{G}_t(x_1, y_1, \ldots, x_t, y_t, w_1, z_1, \ldots, w_{\tau_t}, z_{\tau_t}) .$$

### C.4 From the Online to the Statistical Setting

In this section we show how to utilize our results in the online setting to design an algorithm that achieves exponential rates in the statistical setting. Our approach is similar in spirit as in Bousquet et al. (2021); Hanneke et al. (2022). In particular, we adapt the approach of Hanneke et al. (2022) from the VCL game to the star tree game. The key steps are outlined in Figure 3.

Let P be a realizable distribution with respect to $\mathbb{H}$ and $X_1, Y_1, X_2, Y_2, \ldots$, be an i.i.d. sequence from P. For the remaining of the section, we assume that $\mathbb{H}$ does not have an infinite star tree. Let us now set up the notation that we use in our proof. We denote

$$\tau_t := T_t(X_1, Y_1, \ldots, X_t, Y_t), \quad \widetilde{g}_t(w_1, z_1, \ldots, w_{t_\tau}, z_{t_\tau}) := \widetilde{G}_t(X_1, \ldots, X_t, Y_1, \ldots, Y_t, w_1, z_1, \ldots, w_{t_\tau}, z_{t_\tau}),$$

where these functions are defined in Remark C.3. For any integer $k \geq 1$, we define a "flipping" function $\widetilde{g}_k : (\mathcal{X} \times \{0,1\})^k \to (\mathcal{X} \times \{0,1\})^k$ to be a function that flips one of the labels of its input from 0 to 1 or vice versa. We also define the probability of error of this function as follows:

$$\mathrm{per}(\widetilde{g}_k) = \mathrm{per}^k(\widetilde{g}_k) = \mathrm{P}^{\otimes k}\left((x_1, y_1, \ldots, x_k, y_k) : (x_1, y_1, \ldots, x_k, y_k) \in \mathrm{image}(\widetilde{g}_k)\right) ,$$

i.e., the probability that a $k$-tuple drawn i.i.d. from P belongs to the image of $\widetilde{g}_k$. When there is no confusion about the domain of $\widetilde{g}_k$ we will omit its dependence on $k$. Our first observation is that we get an algorithm that is consistent in the statistical setting. Our proof is an adaptation of Lemma 5.7 in Bousquet et al. (2021).

**Lemma C.4.** *It holds that* $\mathrm{P}[\mathrm{per}(\widetilde{g}_t) > 0] \to 0$ *as* $t \to \infty$.

*Proof.* We know that P is realizable so we can pick a sequence of hypotheses $h_k \in \mathbb{H}$ such that $\mathrm{er}(h_k) \leq 2^{-k}, k \in \mathbb{N}$. Taking a union bound over all $t \geq 1$, we have that

$$\sum_{k \in \mathbb{N}} \mathrm{P}[h_k(X_s) \neq Y_s, \text{for some } s \leq s] \leq t \sum_{k \in \mathbb{N}} \mathrm{er}(h_k) < \infty.$$

Thus, by the Borel-Cantelli lemma, it holds that $h(X_s) = Y_s, \forall s \leq t$ for some $h \in \mathbb{H}$ (with probability one). Let

$$T = \sup\left\{s \geq 1 : (X_{s-\tau_{s-1}+1}, Y_{s-\tau_{s-1}+1}, \ldots, X_s, Y_s) \in \mathrm{image}(\widetilde{g}_{s-1})\right\} .$$

Then, we know from Lemma C.2 that $T$ is finite with probability one. Recall the law of large numbers for $m$-dependent sequences: if $Z_1, Z_2 \ldots$, is an i.i.d. sequence of random variables, then

$$\lim_{n \to \infty} \frac{1}{n} \sum_{i=1}^{n} f(Z_{i+1}, \ldots, Z_{i+m}) = \frac{1}{m} \sum_{i=1}^{m} \lim_{n \to \infty} \frac{m}{n} \sum_{j=0}^{\lfloor n/m \rfloor} f(Z_{mj+1+i}, \ldots, Z_{m(j+1)+i}) + o(1) = \mathbb{E}[f(Z_1, \ldots, Z_m)].$$

Thus, we have that

$$\mathrm{P}[\mathrm{per}^{\tau_t}(\widetilde{g}_t) = 0] = \mathrm{P}\left[\lim_{S \to \infty} \frac{1}{S} \sum_{s=t+1}^{t+S} \mathbb{1}_{(X_s, Y_s, \ldots, X_{s+\tau_t-1}, Y_{s+\tau_t-1}) \in \mathrm{image}(\widetilde{g}_t)} = 0\right]$$

$$\geq \mathrm{P}\left[\lim_{S \to \infty} \frac{1}{S} \sum_{s=t+1}^{t+S} \mathbb{1}_{(X_s, Y_s, \ldots, X_{s+\tau_t-1}, Y_{s+\tau_t-1}) \in \mathrm{image}(\widetilde{g}_t)} = 0, T \leq t\right]$$

$$= \mathrm{P}[T \leq t].$$

We know that $T$ is finite with probability one, so we have that $\mathrm{P}[\mathrm{per}^{\tau_t}(\widetilde{g}_t) > 0] \leq \mathrm{P}[T > t] \to 0$, as $t \to \infty$. $\qquad\square$

Lemma C.4 guarantees that for some $t^* \in \mathbb{N}$, the Gale-Stewart game played on a set of $t^*$ points will have converged with probability at least $7/8$, i.e., $\Pr\left[\mathrm{per}(\widetilde{g}_{t^*}) > 0\right] \leq 1/8$. This number $t^*$ depends on the distribution and is unknown to the learner. We first show how the learner can estimate a similar $\hat{t}_n$ from the data. Our approach is an adaptation of Lemma 5.10 in Bousquet et al. (2021), but we present the proof for completeness.

**Lemma C.5** (Adaptation of Lemma 5.10 in (Bousquet et al., 2021))**.** *For any $N \in \mathbb{N}$, there exists a universally measurable $\hat{t}_N = \hat{t}_n(X_1, Y_1, ..., X_{\lfloor N/2 \rfloor}, Y_{\lfloor N/2 \rfloor})$ whose definition does not depend on* P *so that the following holds. Set the critical time $t^* \in \mathbb{N}$ be such that*

$$\Pr\left[\mathrm{per}(\widetilde{g}_{t^*}) > 0\right] \leq 1/8 ,$$

*where the probability is over the training set of the algorithm. There exist $C, c > 0$ that depend on* P$, t^\star$ *but not $N$ so that*

$$\Pr[\hat{t}_N \in T^\star] \geq 1 - Ce^{-cN} ,$$

*where the probability is over the training of the estimator $\hat{t}_N$ and $T^*$ is the set*

$$T^* = \{1 \leq t \leq t^\star : \Pr\left[\mathrm{per}(\widetilde{g}_t) > 0\right] \leq 3/8\} .$$

*Proof.* We split this labeled set into two parts. The idea is to use the first one to run the Gale-Stewart game and the other one to estimate whether it has converged or not. For each $1 \leq t \leq \lfloor \frac{N}{4} \rfloor$ and $1 \leq i \leq \lfloor \frac{N}{4t} \rfloor$, we let

$$\tau_t^i := T_t(X_{(i-1)t+1}, Y_{(i-1)t+1}, \ldots, X_{it}, Y_{it})$$

$$\widetilde{g}_t^i(w_1, z_1, \ldots, w_{\tau_t^i}, z_{\tau_t^i}) := \widetilde{G}_t(X_{(i-1)t+1}, Y_{(i-1)t+1}, \ldots, X_{it}, Y_{it}, w_1, z_1, \ldots, w_{\tau_t^i}, z_{\tau_t^i}) .$$

be the estimated quantities when we run Figure 5 on the $i$-th batch of the data. Next, for each $t$ we estimate $\Pr[\mathrm{per}(\widetilde{g}_t) > 0]$ by the fraction of the $\widetilde{g}_t^i$ for which there is a tuple on the second quarter of the data that belongs to the image of $\widetilde{g}_t^i$. For every fixed $t$, the data that the functions $\left\{\widetilde{g}_t^i\right\}_{i \leq \lfloor N/2t \rfloor}$ are trained on are independent of each other and of the second half of the training set. This means that we can view every $\left\{\widetilde{g}_t^i\right\}_{i \leq \lfloor N/2t \rfloor}$ as an independent draw of the distribution of $\widetilde{g}_t$. To estimate the error of the algorithm we use the second quarter of the training set. We let

$$\hat{e}_t = \frac{1}{\lfloor N/4t \rfloor} \sum_{i=1}^{\lfloor N/4t \rfloor} \mathbb{1}_{\left\{\exists s: \lfloor N/4 \rfloor + 1 \leq s \leq \lfloor N/2 \rfloor - \tau_t^i, \left(X_{s+1}, Y_{s+1}, \ldots, X_{s+\tau_t^i}, Y_{s+\tau_t^i}\right) \in \mathrm{image}(\widetilde{g}_t^i)\right\}} .$$

Now observe that

$$\hat{e}_t \leq e_t = \frac{1}{\lfloor N/4t \rfloor} \sum_{i=1}^{\lfloor N/4t \rfloor} \mathbb{1}_{\{\mathrm{per}(\widetilde{g}_t) > 0\}} ,$$

with probability one. We define

$$\hat{t}_N = \inf\{t \leq \lfloor N/4 \rfloor : \hat{e}_t < 1/4\} ,$$

where we assume that $\inf \emptyset = \infty$.

We now want to bound the probability that $\hat{t}_N > t^\star$. Using Hoeffding's inequality we get that

$$\Pr\left[\hat{t}_N > t^\star\right] \leq \Pr\left[\hat{e}_{t^\star} \geq \frac{1}{4}\right] \leq \Pr\left[e_{t^\star} \geq \frac{1}{4}\right] = \Pr\left[e_{t^\star} - \frac{1}{8} \geq \frac{1}{8}\right] = \Pr\left[e_{t^\star} - \mathbb{E}[e_{t^\star}] \geq \frac{1}{8}\right] \leq e^{-\lfloor N/4t^\star \rfloor/32} .$$

This implies that $\hat{t}_N \leq t^\star$ except for an event with exponentially small probability. Moreover, for all $1 \leq t \leq t^\star$ that $\Pr\left[\mathrm{per}(\widetilde{g}_t) > 0\right] > \frac{3}{8}$, there is some $\varepsilon > 0$ such that $\Pr\left[\mathrm{per}(\widetilde{g}_t) > \varepsilon\right] > \frac{1}{4} + \frac{1}{16}$ (this holds by continuity). Now fix some $1 \leq t \leq t^\star$ such that $\Pr\left[\mathrm{per}(\widetilde{g}_t) > 0\right] > \frac{3}{8}$ (if it exists). Then, using Hoeffding's inequality again we get that

$$\Pr\left[\frac{1}{\lfloor N/4t \rfloor} \sum_{i=1}^{\lfloor N/4t \rfloor} \mathbb{1}_{\left\{\mathrm{per}(\widetilde{g}_t^i) > \varepsilon\right\}} < \frac{1}{4}\right] \leq e^{-\lfloor N/4t^\star \rfloor/128} .$$

Whenever $g$ is a function such that $\mathrm{per}(g) > \varepsilon$, then

$$\Pr\left[\{\exists s : \lfloor N/4 \rfloor + 1 \leq s \leq \lfloor N/2 \rfloor - \tau, (X_{s+1}, Y_{s+1}, \ldots, X_{s+\tau}, Y_{s+\tau}) \in \mathrm{image}(g)\}\right] \geq 1 - (1 - \varepsilon)^{\lfloor(N-4)/4\tau\rfloor} ,$$

since there are $\lfloor (N-4)/4\tau \rfloor$ disjoint intervals of length $\tau$. As we mentioned before, $\{\widetilde{g}_t^i\}_{i \le \lfloor N/4t \rfloor}$ are independent of $(X_s, Y_s)_{s > \lfloor n/4 \rfloor}$. Thus, applying a union bound we get that the probability that all $\widetilde{g}_t^i$ that have $\mathrm{per}^{\tau_t^i}(\widetilde{g}_t^i) > \varepsilon$ make at least one error on the second half of the training set is

$$
\Pr\left[\mathbb{1}_{\left\{\mathrm{per}^{\tau_t^i}(\widetilde{g}_t^i) > \varepsilon\right\}} \le \mathbb{1}_{\left\{\exists s: \lfloor N/4 \rfloor + 1 \le s \le \lfloor N/2 \rfloor - \tau_t^i, (X_{s+1}, Y_{s+1}, \ldots, X_{s+\tau_t^i}, Y_{s+\tau_t^i}) \in \mathrm{image}(\widetilde{g}_t^i)\right\}}\right]
$$
$$
\ge 1 - \left\lfloor \frac{N}{4t} \right\rfloor (1-\varepsilon)^{\lfloor (N-4)/4t^* \rfloor},
$$

where the last inequality holds since $\tau_t^i \le t^*$. Thus, we get that

$$
\Pr[\hat{t}_N = t] \le \Pr\left[\hat{e}_t < \frac{1}{4}\right] \le \left\lfloor \frac{N}{4} \right\rfloor (1-\varepsilon)^{\lfloor (N-4)/4t^* \rfloor} + e^{-\left\lfloor \frac{N}{4t^*} \right\rfloor / 128}.
$$

Using the previous estimates and applying a union bound, we get that

$$
\Pr[\hat{t}_N \notin T^\star] \le e^{-\lfloor N/4t^* \rfloor / 32} + t^\star \left\lfloor \frac{N}{4} \right\rfloor (1-\varepsilon)^{\lfloor (N-4)/4t^* \rfloor} + t^\star e^{-\lfloor N/4t^* \rfloor / 128} \le C e^{-cn},
$$

for some constants $C, c > 0$. $\qquad\square$

Essentially, Lemma C.5 tells us that we can estimate a batch size $\hat{t}_n$, using only information from the data, so that, with probability $1 - Ce^{-cn}$, the star tree game will have converged when we run it on $\hat{t}$ labeled points. Following the approach of Hanneke et al. (2022), the idea is to run the game multiple times, then create the partial concept classes that are induced by each execution, and, finally, aggregate them into a *majority* class (see Figure 3). We show that if most of the games have converged, then the majority class has bounded star number. This is made formal in Lemma C.6. We remark that Hanneke et al. (2022) showed a similar result regarding the VC dimension of the majority class.

**Lemma C.6.** *The majority class $\mathcal{F}_m^{\hat{N}}$ defined in Figure 3 has star number that is bounded by some distribution-dependent constant $\hat{\mathfrak{s}}$, with probability at least $1 - e^{-c\hat{N}}$, for some absolute constant $c > 0$.*

*Proof.* We know that there exists some $\mathfrak{s}^*$ such that, with probability at least $9/10$, the star of any partial concept class $\mathcal{F}_i$ is at most $\mathfrak{s}^*$. We let $X_i = \mathbb{1}_{\{\text{star number of } \mathcal{F}_i > \mathfrak{s}^*\}}$. Notice the all the $X_i$'s are i.i.d. Bernoulli random variables with $p \le 1/10$, since the data that induce the classes $\mathcal{F}_i$ are i.i.d.. Thus, we can use Hoeffding's inequality to bound the probability that more than $2/10$ of them have star number greater than $\mathfrak{s}^*$ as follows

$$
\Pr\left[\sum_{i=1}^{\hat{N}} X_i \ge (2/10)\hat{N}\right] = \Pr\left[\sum_{i=1}^{\hat{N}} X_i - (1/10)\hat{N} \ge (15/72)\hat{N}\right] \le e^{-\hat{N}/50}.
$$

We let $\mathcal{E}_1$ be the complement of the event above and we condition on it for the rest of the proof. So, we know that at least $8/10\hat{N}$ of the partial concept classes $\mathcal{F}_i$ have star number bounded by $\mathfrak{s}^*$. We will bound the size $\ell$ of the star set of $\mathcal{F}_m^{\hat{N}}$. For any star sequence $S = (x_1, y_1, \ldots, x_\ell, y_\ell) \in (\mathcal{X} \times \{0,1\})^\ell$ of $\mathcal{F}_m^{\hat{N}}$ it holds that

$$
\frac{1}{\ell}\sum_{j=1}^{\ell} \mathbb{1}\left\{\frac{1}{\hat{N}}\sum_{i=1}^{\hat{N}} \mathbb{1}\left\{\exists f \in \mathcal{F}_i : f(x_j) = 1 - y_j, f(x_p) = y_p, \forall p \in [\ell] \setminus \{j\}\right\} > 9/16\right\} = 1.
$$

Using Markov's inequality, we get that

$$
\frac{1}{\ell}\sum_{j=1}^{\ell} \frac{1}{\hat{N}}\sum_{i=1}^{\hat{N}} \mathbb{1}\left\{\exists f \in \mathcal{F}_i : f(x_j) = 1 - y_j, f(x_p) = y_p, \forall p \in [\ell] \setminus \{j\}\right\} > \frac{9}{16}.
$$

Swapping the summation gives us

$$\frac{1}{\hat{N}}\sum_{i=1}^{\hat{N}}\frac{1}{\ell}\sum_{j=1}^{\ell}\mathbb{1}\left\{\exists f\in\mathcal{F}_i:f(x_j)=1-y_j,f(x_p)=y_p,\forall p\in[\ell]\setminus\{j\}\right\}>\frac{9}{16}\iff$$

$$\frac{1}{\hat{N}}\sum_{i=1}^{\hat{N}}\frac{1}{\ell}\sum_{j=1}^{\ell}\mathbb{1}\left\{\nexists f\in\mathcal{F}_i:f(x_j)=1-y_j,f(x_p)=y_p,\forall p\in[\ell]\setminus\{j\}\right\}\le\frac{7}{16}\,.$$

Using Markov's inequality again, we get that

$$\frac{1}{\hat{N}}\sum_{i=1}^{\hat{N}}\mathbb{1}\left\{\frac{1}{\ell}\sum_{j=1}^{\ell}\mathbb{1}\left\{\nexists f\in\mathcal{F}_i:f(x_j)=1-y_j,f(x_p)=y_p,\forall p\in[\ell]\setminus\{j\}\right\}>9/16\right\}\le$$

$$\frac{16}{9}\cdot\frac{1}{\hat{N}}\sum_{i=1}^{\hat{N}}\frac{1}{\ell}\sum_{j=1}^{\ell}\mathbb{1}\left\{\nexists f\in\mathcal{F}_i:f(x_j)=1-y_j,f(x_p)=y_p,\forall p\in[\ell]\setminus\{j\}\right\}\,.$$

This implies that

$$\frac{1}{\hat{N}}\sum_{i=1}^{\hat{N}}\mathbb{1}\left\{\frac{1}{\ell}\sum_{j=1}^{\ell}\mathbb{1}\left\{\nexists f\in\mathcal{F}_i:f(x_j)=1-y_j,f(x_p)=y_p,\forall p\in[\ell]\setminus\{j\}\right\}>9/16\right\}<7/9\iff$$

$$\frac{1}{\hat{N}}\sum_{i=1}^{\hat{N}}\mathbb{1}\left\{\frac{1}{\ell}\sum_{j=1}^{\ell}\mathbb{1}\left\{\exists f\in\mathcal{F}_i:f(x_j)=1-y_j,f(x_p)=y_p,\forall p\in[\ell]\setminus\{j\}\right\}\ge7/16\right\}\ge2/9\,.$$

Thus, we have shown that at least $2/9$ of the partial concept classes witness at least $7/16$ of the star patterns. By the definition of the star number, this means that all these classes have star number at least $7/16\cdot m-1$. This is because we can drop all the points from $S$ that correspond to star patterns that the class does not realize and the remaining set of labeled points is a star set. Since at least one these classes' star number is at most $\mathfrak{s}^*$ we have that $7/16\cdot m-1\le\mathfrak{s}^*$, which means that $m\le16/7(\mathfrak{s}^*+1)$. Thus, we have shown that the star number of $\mathcal{F}_m^{\hat{N}}$ is at most $16/7(\mathfrak{s}^*+1)=\hat{\mathfrak{s}}$, with probability at least $1-e^{-c\hat{N}}$. $\qquad\square$

Similarly as in Hanneke et al. (2022), we define a sequence of universally measurable functions $\left\{G_\ell:(\mathcal{X}\times\{0,1\})^\ell\to\{0,1\}\right\}_{\ell\in\mathbb{N}}$ where

$$G_\ell(x_1,y_1,\ldots,x_\ell,y_\ell)=\mathbb{1}\left\{\exists f\in\mathcal{F}_m^{\hat{N}}:(f(x_1),\ldots,f(x_\ell))=(y_1,\ldots,y_\ell)\right\}\,.$$

An equivalent interpretation of the previous lemma is that the star number of the partial class on which $\{G_\ell\}_{\ell\in\mathbb{N}}$ returns 1 has a distribution-dependent bound, with high probability. Lemma C.7 is an important component in the derivation of our result and it shows that $\{G_\ell\}_{\ell\in\mathbb{N}}$ return 1 on all finite subsets of the correctly labeled data sequence, with high probability. This result follows similarly as Lemma 7 in Hanneke et al. (2022).

**Lemma C.7.** *The $\{G_\ell\}_{\ell\in\mathbb{N}}$ as defined above are universally measurable functions, and with probability at least $1-Ce^{-cN},C,c>0$, the following hold:*

- *The star number of the class*

$$\mathcal{F}_G=\left\{f:\mathcal{X}\to\{0,1,\star\}:\forall\ell\in\mathbb{N},\forall(x_1,\ldots,x_\ell)\in\mathcal{X}^\ell:G_\ell(x_1,f(x_1),\ldots,x_\ell,f(x_\ell))=1\right\}$$

 *has a finite distribution-dependent upper bound $\hat{\mathfrak{s}}$.*

- *$\forall\ell\in\mathbb{N}$, for $(x_1,y_1,\ldots,x_\ell,y_\ell)\sim\mathrm{P}^\ell$, $G_\ell(x_1,y_1,\ldots,x_\ell,y_\ell)=1$ with conditional probability one (given $G_\ell$).*

*Proof.* We condition on the event $\mathcal{E}_0$ described in Lemma C.6. Notice that, by definition, a labeled sequence $S=(x_1,y_1\ldots,x_\ell,y_\ell)\in(\mathcal{X}\times\{0,1\})^\ell$ is a star set of the class $\mathcal{F}_G$ if and only if $S$ is a

star set of $\mathcal{F}_m^{\hat{N}}$. Hence, the bound on the star number follows immediately from Lemma C.6. We also condition on the event $\mathcal{E}_1$ described in Lemma C.5. We now bound the probability that at least $(13/32)\hat{N}$ of the functions $\widetilde{g}_{\hat{t}_N}^i$ have $\mathrm{per}\left(\widetilde{g}_{\hat{t}_N}^i\right) > 0$. For any $t \in T^*$, using Hoeffding's inequality we get that

$$\Pr\left[\frac{1}{\hat{N}}\sum_{i=1}^{\hat{N}}\mathbb{1}\left\{\mathrm{per}\left(\widetilde{g}_{\hat{t}_N}^i\right)>0\right\}>\frac{13}{32}\right] = \Pr\left[\frac{1}{\hat{N}}\sum_{i=1}^{\hat{N}}\mathbb{1}\left\{\mathrm{per}\left(\widetilde{g}_{\hat{t}_N}^i\right)>0\right\}-\frac{3}{8}>\frac{1}{32}\right] \leq \exp\left(-\hat{N}/512\right),$$

Since we know that $\hat{t}_n \leq t^*$, taking a union bound over all $1 \leq t \leq t^*$ we get that

$$\Pr\left[\frac{1}{\hat{N}}\sum_{i=1}^{\hat{N}}\mathbb{1}\left\{\mathrm{per}\left(\widetilde{g}_{\hat{t}_N}^i\right)>0\right\}>\frac{13}{32},\hat{t}_n \in T^*\right] \leq \sum_{t \in T^*}\Pr\left[\frac{1}{\hat{N}}\sum_{i=1}^{\hat{N}}\mathbb{1}\left\{\mathrm{per}\left(\widetilde{g}_{\hat{t}_N}^i\right)>0\right\}>\frac{13}{32}\right]$$
$$\leq t^*\exp\left(-\hat{N}/512\right).$$

Thus, except for an event with exponentially small probability, at least $19/32$ of the functions $\widetilde{g}_{\hat{t}_N}^i$ have $\mathrm{per}\left(\widetilde{g}_{\hat{t}_N}^i\right)$, with probability one. Notice that, by definition, if the partial class $\mathcal{F}_i$ that corresponds to such a function cannot produce a labeling $\vec{y}_\ell \in \{0,1\}^\ell$ of some tuple $\vec{x}_\ell \in \mathcal{X}^\ell$ where $\vec{x}_\ell \sim \mathrm{P}_X^\ell$, we can infer that this $\vec{y}_\ell$ is not the correct labeling, and this inference will be valid with probability one over the draw of $\vec{x}_\ell$. Thus, with probability one, if the $9/16$-majority cannot produce some labeling we have that at least one such partial class $\mathcal{F}_j$ that corresponds to a correct $\widetilde{g}_{\hat{t}_N}^j$ cannot produce this labeling, so it is not the correct one. As a result, with probability one, if $\mathcal{F}_m^{\hat{N}}$ cannot produce $\vec{y}_\ell$ for $\vec{x}_\ell$ we know that this is not the correct labeling. The measurability of $\{G_\ell\}_{\ell \in \mathbb{N}}$ follows by the measurability of all the $\left\{\widetilde{g}_{\hat{t}_N}^i\right\}_{i \in \hat{N}}$. The proof of the lemma follows by noticing that the probability of all the events we have conditioned on can be bounded by $1 - C'e^{-c'\hat{N}}$ and that $\hat{N} = \lfloor N/2\hat{t}_N \rfloor \geq \lfloor N/2t^* \rfloor$. □

We are now ready to prove the result about active learning at exponential universal rates. Our approach is summarized in Figure 4.

**Theorem C.8.** *Assume that $\mathbb{H} \subseteq \{0,1\}^\mathcal{X}$ does not have an infinite star tree. Then, $\mathbb{H}$ admits an active learning algorithm that achieves exponential rates.*

*Proof.* First, notice that the Gale-Stewart subroutine Figure 3 uses $\lfloor n/2 \rfloor$ points and the exponential rates algorithm for partial classes uses at most $\lfloor e^{c \cdot n} \rfloor$ points. This is because we define a uniform distribution over these points during its execution. To prove the learning rate guarantees, we first condition on the event $\mathcal{E}_0$ described in Lemma C.7. Then, we have some partial concept class $\mathcal{F}_G$ whose star number is finite by some distribution-dependent number $\hat{\mathfrak{s}}$ and has the property that $\forall \ell \in \mathbb{N}$, for $(x_1, y_1, \ldots, x_\ell, y_\ell) \sim \mathrm{P}^\ell$, $f(x_1) = y_1, \ldots, f(x_\ell) = y_\ell$ for some $f \in \mathcal{F}_G$. Thus, the conditions of Theorem C.1 are satisfied and the conditional error rate of the output of our algorithm is $\mathbb{E}[\mathrm{er}(\hat{h}_n)|\mathcal{E}_0] \leq c_1\hat{d}e^{-c_2 n/\hat{\mathfrak{s}}}$. Since $\mathcal{E}_0$ happens with probability at least $1 - \tilde{C}e^{-\tilde{c}n}$, we see that the unconditional error of the output is $\mathbb{E}[\mathrm{er}(\hat{h}_n)] \leq c_1\hat{d}e^{-C_2 n/\hat{\mathfrak{s}}} + \tilde{C}e^{-\tilde{c}n} \leq Ce^{-cn}$, for some distribution-dependent constants $c, C > 0$. □

## C.5 Sublinear Rates Lower Bound

**Theorem C.9.** *Assume that $\mathbb{H} \subseteq \{0,1\}^\mathcal{X}$ has an infinite star tree and let $R(n)$ be a rate function with $R(n) = o(1/n)$. Then, $\mathbb{H}$ requires rate $R(n)$.*

*Proof.* Let

$$\vec{t} = \bigcup_{0 \leq \ell < \infty}\left\{(x_{\vec{u}}, z_{\vec{u}}) \in \left(\mathcal{X}^{\ell+1}, \{0,1\}^{\ell+1}\right) : \vec{u} \in \{0\} \times \{0,1\} \times \ldots \times \{0,1,\ldots,\ell\}\right\},$$

be an infinite star tree. We also fix the learning algorithm $\hat{h}_n$. Recall that $\{u(n)\}_{n\in\mathbb{N}}$ denotes the number of unlabeled samples that $\hat{h}_n$ uses.

We first pick a path on the tree uniformly at random. To be more precise, let $\vec{y} = (y_1, y_2, \dots)$ be a sequence of independent random variables where $y_i$ is uniformly distributed in $\{0, \dots, i-1\}$. The idea is that $\vec{y}$ specifies a random path on the tree. We define the target distribution $\mathrm{P}_{\vec{y}}$ in an inductive way by specifying sequences $\{k_i, p_i, n_i\}_{i\geq 1}$ and putting mass $p_i$ on the node of the random path at level $k_i$. Conditional on the given node, we distribute the mass among its points uniformly. Let us now be formal. For the base of the induction, we let $p_2 = 1/2$ and $k_1 = 1$. We will specify the value of $p_1$ later. For the inductive step, for any $i \geq 2$, given the value of $p_i, k_{i-1}$, we will specify $k_i, p_{i+1}$. We let $k_i$ be the smallest integer so that $k_i > k_{i-1}$ and $R(\lfloor k_i/8 \rfloor) < p_i/k_i$. Notice that since $R(n) = o(1/n)$, $k_i < \infty$. We let $n_i = \lfloor k_i/8 \rfloor$ and

$$p_{i+1} = \min\left\{\frac{1}{4u_{n_i}}, \frac{p_i}{4}\right\} .$$

Notice that since $\{p_i > 0\}_{i\geq 2}, \sum_{i\geq 2} p_i < 1^4$, by setting $p_1 = 1 - \sum_{i\geq 2} p_i$ the sequence $\{p_i\}_{i\geq 1}$ can be used as a probability distribution. We are now ready to define $\mathrm{P}_{\vec{y}}$ on $\mathcal{X} \times \{0,1\}$. To make the notation simpler, for any $\vec{y}$ and for $k \geq 1$

$$w^j_{\vec{y}\leq k-1} = \begin{cases} 1 - z^j_{\vec{y}\leq k-1}, & \text{if } j = y_k \\ z^j_{\vec{y}\leq k-1}, & \text{otherwise} \end{cases} .$$

For $i \geq 1$, let

$$\mathrm{P}_{\vec{y}}\left(x^j_{\vec{y}\leq k_i-1}, w^j_{\vec{y}\leq k_i-1}\right) = \frac{p_i}{k_i}, 1 \leq j \leq k_i - 1 .$$

Notice that, by the definition of $p_i$, this is indeed a probability distribution. We now show that $\mathrm{P}_{\vec{y}}$ is realizable. Since $\vec{t}$ is a star tree, there exists some $h \in \mathbb{H}$ such that

$$h\left(x^j_{\vec{y}\leq k}\right) = w^j_{\vec{y}\leq k-1} ,$$

for all $0 \leq j \leq k$, and $1 \leq k \leq n$. Thus, for the probability of error of $h$ we have that

$$\mathrm{er}_{\vec{y}}(h) := \mathrm{P}_{\vec{y}}\left[(x,y) \in \mathcal{X} \times \{0,1\} : h(x) \neq y\right] \leq \sum_{k>n} p_k ,$$

so by taking $n \to \infty$ we see that $\mathrm{P}_{\vec{y}}$ is realizable for every $\vec{y}$. Moreover, we note that the mapping $\vec{y} \to \mathrm{P}_{\vec{y}}$ is measurable.

We now let $(X, Y), (X_1, Y_1), (X_2, Y_2), \dots$ be i.i.d. samples drawn from $\mathrm{P}_{\vec{y}}$. We can draw them as

$$X = x^J_{\vec{y}\leq T-1}, \quad Y = w^J_{\vec{y}\leq T-1}, \quad X_j = x^{J_j}_{\vec{y}\leq T_j-1}, \quad Y_j = w^{J_j}_{\vec{y}\leq T_j-1} ,$$

where $(T, J), (T_1, J_1), (T_2, J_2), \dots$ are i.i.d. random variables, independent of the path $\vec{y}$, with

$$\Pr[T = k_i, J = j] = \frac{p_i}{k_i} ,$$

for all $i \geq 1, 0 \leq j \leq k_i - 1$.

For all $i \in \mathbb{N}$, let us define the event $E^i_0 = \left\{T_1 \leq k_i, T_2 \leq k_i, \dots, T_{u_{n_i}} \leq k_i\right\}$ under which the algorithm has not received any unlabeled points from any level $k_j, j > i$. Let us also call $E^i_1$ the event that $\hat{h}_{n_i}$ does not query the point whose label is flipped, i.e., the point $x^{y_{k_i}}_{\vec{y}\leq k_i-1}$. Also, we denote $E^i_2$ the event that the learner classifies at least one point of level $k_i$ incorrectly. Then, we have that

$$\Pr[E^i_2|E^i_1, E^i_0] \geq \left(1 - \frac{1}{(7/8 \cdot k_i)}\right) \geq \frac{6}{7} ,$$

since there are at least $(7/8) \cdot k_i$ points on this level that the learner has not queried, and under the events we have conditioned on the flipped point along the path is chosen uniformly at random among these points. We now bound $\Pr[E^i_0]$. Notice that

$$\sum_{j\geq i+1} p_j \leq \frac{4}{3}p_{i+1} \leq \frac{p_i}{3} \leq \frac{1}{12 \cdot u_{n_i}} .$$

---

[4]This is because $p_{i+1} \leq p_i/4$.

Thus,

$$\Pr[E_0^i] \geq \left(1 - \sum_{j \geq i+1} p_j\right)^{u_{n_i}} \geq \left(1 - \frac{1}{12 \cdot u_{n_i}}\right)^{u_{n_i}} \geq \frac{9}{10}\,.$$

Finally, we need to bound $\Pr[E_1^i | E_0^i]$. Notice that

$$\Pr[E_1^i | E_0^i] \geq \frac{7}{8}\,,$$

since the learner has $n$ label queries and there are at least $8n$ points. Putting it together, we can bound

$$\Pr[E_2^i] = \Pr[E_2^i | E_1^i, E_0^i] \cdot \Pr[E_1^i | E_0^i] \cdot \Pr[E_0^i]$$
$$\geq \frac{6}{7} \cdot \frac{7}{8} \cdot \frac{9}{10}$$
$$= \frac{27}{40}\,,$$

which using reverse Markov's inequality implies that

$$\Pr_{\vec{y}}\left[\Pr[E_2^i | \vec{y}] \geq \frac{9}{40}\right] \geq \frac{18}{31}\,.$$

Moreover, notice that

$$\Pr[\hat{h}_{n_i}(X) \neq Y | \vec{y}] \geq \frac{p_i}{k_i} \cdot \Pr[E_2^i | \vec{y}] \geq R(n_i) \cdot \Pr[E_2^i | \vec{y}]\,.$$

Thus, we can see that

$$\Pr_{\vec{y}}\left[\Pr[\hat{h}_{n_i} \neq Y | \vec{y}] \geq \frac{9}{40} \cdot R(n_i)\right] \geq \frac{18}{31}\,.$$

Using Fatou's lemma, we have that

$$\mathbb{E}\left[\limsup_{i \to \infty} \frac{1}{R(n_i)} \Pr[\hat{h}_{n_i}(X) \neq Y | \vec{y}]\right] \geq \mathbb{E}\left[\limsup_{i \to \infty} \frac{1}{R(n_i)} \cdot \min\left\{\Pr[\hat{h}_{n_i}(X) \neq Y | \vec{y}], 9/40 \cdot R(n_i)\right\}\right]$$
$$\geq \limsup_{i \to \infty} \frac{1}{R(n_i)} \mathbb{E}\left[\min\left\{\Pr[\hat{h}_{n_i}(X) \neq Y | \vec{y}], 9/40 \cdot R(n_i)\right\}\right]$$
$$\geq \limsup_{i \to \infty} \frac{1}{R(n_i)} \Pr_{\vec{y}}\left[\Pr[\hat{h}_{n_i}(X) \neq Y | \vec{y}] \geq 9/40 \cdot R(n_i)\right] \cdot \frac{9}{40} \cdot R(n_i)$$
$$\geq \frac{9}{40} \cdot \frac{18}{31}$$
$$= \frac{81}{620}\,,$$

where the first inequality is by the definition of $\min$, the second inequality follows from Fatou's lemma, and the third inequality follows from Markov's inequality. We remark that Fatou's lemma can be applied here since

$$\frac{1}{R(n_i)} \min\left\{\Pr\left[\hat{h}_{n_i}(X) \neq Y | \vec{y}\right], 9/40 \cdot R(n_i)\right\} \leq 9/40\,.$$

Thus, there must exist a realization of $\vec{y}$ such that

$$\mathbb{E}[\mathrm{er}_{\vec{y}}(\hat{h}_n) | \vec{y}] \geq C \cdot R(n)\,,$$

infinitely often, where $C$ is some absolute constant. Choosing $\mathrm{P} = \mathrm{P}_{\vec{y}}$ for this realization of $\vec{y}$ concludes the proof. $\qquad\square$

## D    Omitted Details from Section 4

### D.1    Useful Tools

**Theorem D.1** (Dvoretzky–Kiefer–Wolfowitz (DKW) Inequality)**.** *Let $F$ be the CDF of some distribution $\mathcal{D}$ supported on $\mathbb{R}$, $n \in \mathbb{N}$ and $F_n$ be the empirical CDF obtained from $n$ i.i.d. samples from $\mathcal{D}$. Then, $\forall \varepsilon > 0$ we have that*

$$\Pr\left[\sup_{x \in \mathbb{R}} |F_n(x) - F(x)| > \varepsilon\right] \leq 2e^{-2n\varepsilon^2}\,.$$

Figure 6: The VCL Game (Bousquet et al., 2021).

**VCL Game Subroutine (Bousquet et al., 2021):** Input is a labeled sequence
$\{(x_1, y_1), \ldots, (x_N, y_N)\}$
    1. Let $\tau_0 \leftarrow 0$.
    2. In every step $t \geq 1$ :
        • If $\eta_{\tau_{t-1}}(\xi_1, \ldots, \xi_{\tau_{t-1}-1}, x_{t-\tau_{t-1}+1}, \ldots, x_t) = (y_{t-\tau_{t-1}+1}, \ldots, y_t)$ :
          – Let $\xi_{\tau_{t-1}} \leftarrow (x_{t-\tau_{t-1}+1}, \ldots, x_t)$ and $\tau_t \leftarrow \tau_{t-1} + 1$.
        • Else, $\tau_t \leftarrow \tau_{t-1}$.

## D.2 The VCL Game

For completeness, we provide various algorithms and results that appeared in Bousquet et al. (2021); Hanneke et al. (2022), which we utilize in our approach to get sublinear learning rates. It is instructive to start with an informal description of a pattern avoidance function $g : \mathcal{X}^k \to \{0, 1\}^k$. Intuitively, this function takes as input a $k$-tuple of unlabeled data generated by a realizable distribution $P_\mathcal{X}$ and it returns a labeling of these data that is not the true labeling by P. They define

$$\mathrm{per}(g) = P^k[x_1, \ldots, x_k : g(x_1, \ldots, x_k) = (y_1, \ldots, y_k)].$$

Bousquet et al. (2021) provide a way to obtain such a pattern avoidance function for any realizable distribution. This is obtained by a VCL game in Figure 6 which returns a function $\hat{\vec{y}}_{\tau_t}$. Their following result shows that as the number of points $N$ used in the VCL game grows, the probability that we obtain a pattern avoidance function that is incorrect goes to zero.

**Lemma D.2** (Correct Pattern Avoidance Function (Restatement of Lemma 5.7 (Bousquet et al., 2021))). *Consider the VCL game in Figure 6. Then, $\Pr[\mathrm{per}(\hat{\vec{y}}_{\tau_t}) > 0] \to 0$, as $N \to \infty$.*

Finally, we remark that Bousquet et al. (2021) showed the following result.

**Theorem D.3** (Supervised Learning Linear Rates (Bousquet et al., 2021)). *If $\mathbb{H}$ does not have an infinite VCL tree there exists a supervised learning algorithm that can learn $\mathbb{H}$ at linear rate.*

## D.3 Omitted Details of the Sublinear Rates Algorithm

**Lemma D.4.** *Let $\mathcal{F}^i_{\sqrt{n}/5}, i \in [\sqrt{n}]$ be the partial concept classes that are obtained by playing the VCL game on $\sqrt{n}/5$ examples that are generated i.i.d. from P. Then, with probability at least $1 - e^{-C \cdot n^{1/4}}$, where $C > 0$ is some absolute constant, P is a realizable distribution for a $(1 - o(1))$-fraction of these classes.*

*Proof.* Let $\widehat{\widetilde{y}}^i_{\sqrt{n}/5}$ be the pattern avoidance function that is obtained from the $i$-th batch of $\sqrt{n}/5$ i.i.d. samples from P. Since we know that $\Pr[\mathrm{per}(\widehat{\widetilde{y}}^i_{\sqrt{n}/5}) > 0] \to 0$ as $n \to \infty$ (cf. Lemma D.2), for any $\varepsilon > 0$ there is some $b_\varepsilon$ such that if the batch size $b = \sqrt{n}/5$ is at least $b_\varepsilon$, then $\Pr[\mathrm{per}(\widehat{\widetilde{y}}^i_{\sqrt{n}/5}) > 0] < \varepsilon$. Hence, for every $n$ there is some $\varepsilon_n = o(1)$ so that when the batch size is $\sqrt{n}/5$ we have that $\Pr[\mathrm{per}(\widehat{\widetilde{y}}^i_{\sqrt{n}/5}) > 0] < \varepsilon_n$. Notice that whenever $\mathrm{per}(\widehat{\widetilde{y}}^i_{\sqrt{n}/5}) = 0$ the distribution P is realizable with respect to the corresponding class $\mathcal{F}^i_{\sqrt{n}/5}$. Let us set $\delta_n = n^{-1/8}$. Using Hoeffding's inequality we have that with probability at least $1 - e^{-C \cdot \delta_n^2 \cdot \sqrt{n}} = 1 - e^{C \cdot n^{1/4}}$, where $C$ is some absolute constant, at least $(1 - \varepsilon_n - \delta_n)$-fraction of the $\widehat{\widetilde{y}}^i_{\sqrt{n}/5}$ will have $\mathrm{per}(\widehat{\widetilde{y}}^i_{\sqrt{n}/5}) = 0$. Since $\delta_n + \varepsilon_n = o(1)$ we have shown the claim. $\square$

**Lemma D.5.** *Let $\widetilde{P}$ be the distribution of the VC dimension of the partial concept classes that are obtained by playing the VCL game on infinitely many i.i.d. samples from P. Let $d^*$ be the $9/10$-quantile of $\widetilde{P}$. Then, for $n \geq n_P$, where $n_P$ is some P-dependent constant we have that $\widehat{d}_n = d^*$, with probability at least $1 - 3e^{-C_P \cdot n^{1/4}}$, where $C_P$ is a constant the depends on the data-generating distribution P.*

*Proof.* Consider the following experiment. We sample an infinite sequence of labeled points from P and we run the VCL game (cf. Figure 6) on this sequence. Let $\widehat{y}$ be the pattern avoidance function that is obtained from that game defined over $\widehat{\ell}$ many points and let

$$\widehat{\mathcal{F}} := \left\{ f : \mathcal{X} \to \{0, 1, \star\} : (f(x_1), \ldots, f(x_{\widehat{\ell}})) \neq \widehat{y}(x_1, \ldots, x_{\widehat{\ell}}) \right\} .$$

Let $d^* \in \mathbb{N}$ be the smallest number such that

$$\Pr[\mathrm{d}(\widehat{\mathcal{F}}) \leq d^*] > 9/10 .$$

Let $F$ be the CDF of the distribution of the VC dimension of $\mathcal{F}$, let $d_1 < d^* < d_2$ be the largest, smallest number[5] such that $F(d_1) \neq F(d^*), F(d_2) \neq F(d^*)$ and let $C_P = \min\{F(d^*) - F(d_1), F(d_2) - F(d^*)\}/4$. Let $F_{\sqrt{n}}$ be the empirical CDF of the distribution obtained on $\sqrt{n}$ i.i.d. samples. Notice that whenever we estimate $F$ with accuracy $C_P$ the empirical $9/10$-quantile will the same as the true $9/10$-quantile. The DKW inequality (Theorem D.1) shows that this happens with probability $1 - 2e^{-2C_P^2 k}$. Let us call this even $E_1$ and condition on it.

Next, we need to handle the fact that we do not obtain samples from partial concept classes that are generated from VCL games on infinitely many i.i.d. samples from P but merely on $\sqrt{n}$ many such samples. Let us consider the following experiment. We run the VCL game on $k$ streams of infinitely many i.i.d. samples from P and if the game has not terminated within the first $\sqrt{n}$ rounds, we rewind the pattern avoidance function to the one that is obtained in that round. Then, Lemma D.4 shows that, with probability at least $1 - e^{-C \cdot n^{1/4}}$, we will only rewind the pattern avoidance function of a $o(1)-$fraction of these games will have converged. We call this event $E_2$ and we condition on it. Let us denote by $\widehat{F}_k$ the empirical CDF of the VC dimension of the partial concept classes obtained by this experiment. Under $E_2$, we have that $\sup_{x \in \mathbb{N}} |F_k(x) - \widehat{F}_k(x)| = o(1)$. Thus, by taking $n$ large enough so that the $o(1) << C_P$, we see that the estimation of $\widehat{\mathrm{d}}_n$ using the samples that are obtained from the truncated VCL game also converges to $\mathrm{d}^*$. The bound on the probability of error follows from a union bound over $E_1, E_2$. $\qquad\square$

**Proposition D.6.** *For $n > n_P$, where $n_P$ is some distribution-dependent constant, given $m = O(n^3)$ unlabeled samples we can estimate $p_{n,k}, p_{n,k}^X, p_{n,k}^{(X,y)}$ with accuracy $o(1)$, with probability at least $1 - C_P \cdot e^{-C_P' n^{1/4}}$, where $C_P, C_P' > 0$, are distribution-dependent constants.*

*Proof.* As we alluded to before, using $O(n^3)$ unlabeled samples in total we can estimate the quantities $p_{n,k,i}, p_{n,k,i}^X, p_{n,k,i}^{(X,y)}$ with accuracy $1/n$ for each version space $V_{n/5}^i$, with probability at least $1 - C_1 \sqrt{n} e^{-C_2 \sqrt{n}}$, where $C_1, C_2 > 0$, are absolute constants. Let us denote these estimates $\widehat{p}_{n,k,i}, \widehat{p}_{n,k,i}^X, \widehat{p}_{n,k,i}^{(X,y)}$ We condition on this event for the rest of the proof.

Assume that $n$ is large enough and condition on the events of Lemma D.4, Lemma D.5. Let us denote by $i_1, \ldots, i_k$ the indices of the partial concept classes whose VC dimension is bounded by $\widehat{\mathrm{d}}_n = \mathrm{d}^*$. By definition, $k \geq 9/10\sqrt{n}$. Consider the following statistical experiment. We run the VCL game on infinitely many i.i.d. samples from P, we define the partial concept class obtained from the pattern avoidance function of the game, if the class has VC dimension at most $\mathrm{d}^*$ we keep the sample, otherwise we discard it. Then, we define its version space on the first $n/5$ samples of $S_2^\infty$. Let us call $\widetilde{V}^i$ the version space obtained by the $i$-th sample of this experiment. Then, $P^k(x_1, \ldots, x_k$ VC shattered by $\widetilde{V}^i)$ is an unbiased sample from the distribution whose expected value $p_{n,k}$ we are trying to estimate. Similarly for the rest of the quantities we have defined. Let us call these samples $\widetilde{p}_{n,k,i}, \widetilde{p}_{n,k,i}^X, \widetilde{p}_{n,k,i}^{(X,y)}$ and denote by $\widetilde{S}_n, \widetilde{S}_n^X, \widetilde{S}_n^{(X,y)}$ their empirical estimates over $9/10\sqrt{n}$ many samples. Hoeffding's inequality gives us that $|\widetilde{S}_n - p_{n,k}| = o(1), |\widetilde{S}_n^X - p_{n,k}^X| = o(1), |\widetilde{S}_n^{(X,y)} - p_{n,k}^{(X,y)}| = o(1)$, with probability at least $1 - Ce^{-C' \cdot n^{1/4}}$, for some absolute constants $C, C' > 0$. We condition on this event.

Notice that, under the events we have conditioned on, the samples $p_{n,k,i}, p_{n,k,i}^X, p_{n,k,i}^{(X,y)}$ we obtain from running the VCL game on $O(\sqrt{n})$ many samples, differ from the samples $\widetilde{p}_{n,k,i}, \widetilde{p}_{n,k,i}^X, \widetilde{p}_{n,k,i}^{(X,y)}$

---

[5]If only one these number exists, then we ignore the one that does not exist in the definition of $C_P$. If none of these numbers exist then the estimation task is trivial.

on a sublinear number of terms. Moreover, the absolute difference on the terms they disagree on is bounded by 1. Thus, if we denote by $S_n, S_n^X, S_n^{(X,y)}$ the average of these values we have that $|\widetilde{S}_n - S_n| = o(1), |\widetilde{S}_n^X - S_n^X| = o(1), |\widetilde{S}_n^{(X,y)} - S_n^X| = o(1)$. Finally, let $\widehat{S}_n, \widehat{S}_n^X, \widehat{S}_n^{(X,y)}$ be the averages of $\widehat{p}_{n,k,i}, \widehat{p}_{n,k,i}^X, \widehat{p}_{n,k,i}^{(X,y)}$. Since $|\widehat{p}_{n,k,i} - p_{n,k,i}| = o(1), |\widehat{p}_{n,k,i}^X - p_{n,k,i}^X| = o(1), |\widehat{p}_{n,k,i}^{(X,y)} - p_{n,k,i}^{(X,y)}| = o(1)$, using the triangle inequality on all the previous estimations we see that $|\widehat{S}_n - p_{n,k}| = o(1), |\widehat{S}_n^X - p_{n,k}^X| = o(1), |\widehat{S}_n^{(X,y)} - p_{n,k}^{(X,y)}| = o(1)$, with probability at least $1 - C_P \cdot e^{-C_P' n^{1/4}}$. $\quad\square$

**Lemma D.7.** *Let $n > C_{S_2^\infty}$, where $C_{S_2^\infty}$ is some constant that depends on $S_2^\infty, P$. Then, with probability at least $1 - C_P \cdot e^{-C_P' n^{1/4}}$, for every $k \leq k^*$, if $\widehat{p}_{n,k}^X < \frac{\widehat{p}_{n,k}}{2}$ then $y = \mathrm{argmax}_{y \in \{0,1\}} \widehat{p}_{n,k}^{(X,y)}$ is the correct label of $X$.*

*Proof.* Let $n > C_{S_2^\infty}$ be large enough so that for all $k \leq k^*$ we have that:

- 
$$p_{n,k} \leq 100/99 \cdot \lim_{n \to \infty} p_{n,k},$$

- the guarantee from Proposition D.6 gives us that

$$\widehat{p}_{n,k}^X < \frac{\widehat{p}_{n,k}}{2} \implies p_{n,k}^X < \frac{3}{4} \cdot p_{n,k},$$

- and

$$p_{n,k}^{(X,y)} \geq \frac{99}{100} p_{n,k} \implies \widehat{p}_{n,k}^{(X,y)} \geq \frac{95}{100} \widehat{p}_{n,k}$$

$$p_{n,k}^{(X,y)} < \frac{80}{100} p_{n,k} \implies \widehat{p}_{n,k}^{(X,y)} < \frac{90}{100} \widehat{p}_{n,k}.$$

We condition on these events for the rest of the proof. Since $p_{n,k} \leq 100/99 \cdot \lim_{n \to \infty} p_{n,k}$ for the correct label $y^*$ we have that $p_{n,k}^{(X,y^*)} \geq 99/100 \cdot p_{n,k}$. Moreover, since $p_{n,k}^X < 3/4 \cdot p_{n,k}$ for the other label $\bar{y} = 1 - y^*$ it holds that $p_{n,k}^{(X,\bar{y})} < 80/100 \cdot p_{n,k}$. Thus, under the event we have conditioned on we have that $\mathrm{argmax}_{y \in \{0,1\}} \widehat{p}_{n,k}^{(X,y)}$ will indeed be the correct label. The correctness probability follows directly from Proposition D.6. $\quad\square$

**Lemma D.8.** *For $k = k^*$ we have that $\Pr\left[\widehat{p}_{n,k^*}^X \geq \frac{\widehat{p}_{n,k^*}}{2}\right] = o(1)$.*

*Proof.* Essentially, the proof of this result follows by Markov's inequality. Let us consider some large enough $n$ and condition on the event described in Proposition D.6. Then, $\widehat{p}_{n,k^*}^X \geq \frac{\widehat{p}_{n,k^*}}{2} \implies p_{n,k^*}^X \geq \frac{p_{n,k^*}}{4}$, so it suffices to bound

$$\Pr\left[p_{n,k^*}^X \geq \frac{p_{n,k^*}}{4}\right].$$

Using Markov's inequality, we have that

$$\Pr\left[p_{n,k^*}^X \geq \frac{p_{n,k^*}}{4}\right] \leq \frac{4 \, \mathbb{E}[p_{n,k^*}^X]}{p_{n,k^*}}$$

$$= \frac{4 p_{n,k^*+1}}{p_{n,k^*}}$$

$$= o(1).$$

where we have used the fact that $\mathbb{E}[p_{n,k^*}^X] = p_{n,k^*+1}$ and that $p_{n,k^*}$ is bounded below by a constant.

$\quad\square$

We now state the guarantees of ActiveSelect (Hanneke, 2012). We note that the original statement of the result did not specify the size of the unlabeled dataset that the algorithm uses, but a straightforward adaptation of it shows that with $O(n^3)$ many unlabeled data its error guarantee changes increases by $O(1/n^2)$.

**Lemma D.9** (Hanneke (2012))**.** *A call to ActiveSelect (cf Figure 8) with $N$ classifiers $\{h_1, \ldots, h_N\}$, a query budget of $m$, and $O(N^2/n^3)$ unlabeled points makes at most $m$ label queries and if $h_{\widehat{k}}$ is the classifier it returns, then with probability at least $1 - C_1 \cdot N e^{-C_2 \cdot m/(N^2 \log(N))}$, we have $\mathrm{er}(h_{\widehat{k}}) \leq 2 \min_{k \in [N]} \mathrm{er}(h_k) + O(1/n^2)$.*

We are now ready to prove the main result in this section. The algorithm is summarized in Figure 7.

**Theorem D.10.** *Assume that $\mathbb{H} \subseteq \{0,1\}^{\mathcal{X}}$ does not have an infinite VCL tree. Then, $\mathbb{H}$ admits an active learning algorithm that achieves sublinear rates.*

*Proof.* We have all the main ingredients we need to prove our result. Let us fix some $S_2^{\infty}$ whose elements are i.i.d. from P. Conditioning on the results of Lemma D.7, Lemma D.8, we have that for large enough $n$, which depends on $S_2^{\infty}$, we can obtain a labeled dataset $L_n$ whose size is $\omega(n)$. Feeding this dataset into the supervised learning algorithm from Bousquet et al. (2021) gives as a classifier $\widetilde{h}_n$ that, conditioned on $S_2^{\infty}$ and the events $E$ we have described, has error

$$\mathbb{E}[\mathrm{er}(\widetilde{h}_n)|S_2^{\infty}, E] = o(1/n)\,.$$

Moreover, the probability of $E$ is also $o(1/n)$, so we get

$$\mathbb{E}[\mathrm{er}(\widetilde{h}_n)|S_2^{\infty}] = o(1/n)\,.$$

Similarly, for the output $\widehat{h}_n$ of ActiveSelect, we have that

$$\mathbb{E}[\mathrm{er}(\widehat{h}_n)|S_2^{\infty}] = o(1/n)$$
$$\mathbb{E}[\mathrm{er}(\widehat{h}_n)] = O(1/n)\,,$$

where the second bound comes from the fact that we feed into ActiveSelect the output of the passive learning algorithm $\widehat{h}_n^p$ from Bousquet et al. (2021) (cf. Theorem D.3), whose bound does not not depend on $S_2^{\infty}$.

Let us call $R(n)$ the error rate of our algorithm. To show that $\mathbb{E}[R(n)] = o(1/n)$ it suffices to prove that $\limsup_{n \to \infty} \mathbb{E}[n \cdot R(n)] = 0$. Fatou's lemma gives us that

$$\begin{aligned}
\limsup_{n \to \infty} n \cdot \mathbb{E}[\cdot R(n)] &= \limsup_{n \to \infty} n \cdot \mathbb{E}[\cdot R(n)] \\
&= \limsup_{n \to \infty} \mathbb{E}_{S_2^{\infty}} [\mathbb{E}[n \cdot R(n)|S_2^{\infty}]] \\
&\leq \mathbb{E}_{S_2^{\infty}} [\limsup_{n \to \infty} n \cdot \mathbb{E}[R(n)|S_2^{\infty}]] \\
&= 0\,,
\end{aligned}$$

where Fatou's lemma applies since $\mathbb{E}[n \cdot R(n)|S_2^{\infty}] \leq C_{\mathrm{P}}$ where $C_{\mathrm{P}}$ is some P$-$dependent constant. This is because, almost surely over $S_2^{\infty}$, the expected error of $\widehat{h}_n^p$ is bounded by $C'_{\mathrm{P}}/n$.

$\square$

## D.4 Arbitrarily Slow Rates Lower Bound

**Theorem D.11** (Hanneke et al. (2022))**.** *If $\mathbb{H}$ has an infinite VCL tree, no active learning algorithm can achieve rates that are faster than arbitrarily slow.*

## D.5 Omitted Figures

Figure 7: Sublinear Rates Algorithm.

**Sublinear Rates Algorithm** [Adaptation from Hanneke (2012)] Input is a label budget $N$ and a passive learning algorithm $A_p$

1. Let $S_1, S_2, S_3, S_4, S_5, U$ be sets of $O(N^3)$ unlabeled points that are i.i.d. from $\mathrm{P}_{\mathcal{X}}$.

2. Request the labels of the first $N/5$ points of $S_5$ and run $A_p$ on this set. Let $\widehat{h}_N^p$ be its output.

3. Request the labels of the first $N/5$ points of $S_1$, split them into batches of size $O(\sqrt{N})$ and run the VCL game on them (cf. Figure 6).

4. Let $\widehat{y}^i_{\sqrt{N}/t}, i \in [\sqrt{N}]$, be the pattern avoidance functions from the previous step and for every $i \in [\sqrt{N}]$ define

$$\mathcal{F}^i_{\sqrt{N}/5} := \left\{ f : \mathcal{X} \to \{0, 1, \star\} : (f(x_1), \ldots, f(x_{\ell^i_{\sqrt{N}/5}})) \neq \right.$$

$$\left. \widehat{y}^i_{\sqrt{N}/5}(x_1, \ldots, x_{\ell^i_{\sqrt{N}/5}}) \right\}.$$

5. Let

$$\widehat{\mathrm{d}}_N := \min_{d \in \mathbb{N}} \left\{ \exists i_1, i_2, \ldots, i_{9/10 \cdot \sqrt{N}} \in [\sqrt{N}] : \right.$$

$$\left. i_1 < i_2 < \ldots < i_{9/10 \cdot \sqrt{N}}, \mathrm{d}(\mathcal{F}^{i_j}_{\sqrt{N}/5}) \leq d, \forall j \in [9/10\sqrt{N}] \right\}.$$

6. Request the labels of the first $N/5$ points of $S_2$ and let

$$V^i_{N/5} := \left\{ f \in \mathcal{F}^i_{\sqrt{N}/5} : f(x) = y \text{ for these } N/5 \text{ point} \right\}.$$

7. For $k = 0, \ldots, \widehat{\mathrm{d}}_N$:

   (a) Let $\widetilde{S}_3$ be the next $n^2$ examples in $S_3$ and $\widetilde{m}_N = N/(5(\widehat{d}_N + 1))$ be the label budget.

   (b) Let $\widehat{S} = \{\}$

   (c) For each point $X \in \widetilde{S}_3, y \in \{0, 1\}$:

      i. Estimate $\widehat{p}_{N,k}, \widehat{p}^X_{N,k}, \widehat{p}^{(X,y)}_{N,k}$ using $V^i_{N/5}$ and fresh unlabeled points from $U$ as described in Proposition D.6.

      ii. If $\widehat{p}^X_{N,k} \leq \widehat{p}_{N,k}/2$ infer $y^* = \mathrm{argmax}_{y \in \{0,1\}} \widehat{p}^{(X,y)}_{N,k}$, otherwise request the label $y^*$ of $X$.

      iii. Append $(X, y^*)$ to $\widehat{S}$ and update the label budget $\widetilde{m}_N$. If $\widetilde{m}_N = 0$ jump to Step (d).

   (d) Run $A_p$ on $\widehat{S}$ and let $\widehat{h}_k$ be its output.

8. Return $\mathrm{ActiveSelect}\left( \left\{ \widehat{h}_0, \ldots, \widehat{h}_k, \widehat{h}_N^p \right\}, N/5, S_4 \right)$ (cf. Figure 8).

Figure 8: ActiveSelect Algorithm (Hanneke, 2012).

**ActiveSelect** (Hanneke, 2012) Input is a set of classifiers $\{h_1, \ldots, h_L\}$, label budget $m$, sequence of unlabeled examples $\mathcal{U}$.

    1. For each $j, k \in \{1, \ldots, N\}$ s.t. $j < k$:

        (a) Let $R_{jk}$ be the first $\left\lfloor \frac{m}{j(L-j)\ln(eL)} \right\rfloor$ points in $\mathcal{U} \cap \{x : h_j(x) \neq h_k(x)\}$ (if such value exists).

        (b) Request the labels for $R_{jk}$ and let $Q_{jk}$ be the set of labeled examples.

        (c) Let $m_{kj} = \mathrm{er}_{Q_{jk}}(h_k)$.

    2. Return $h_{\hat{k}}$, where $\hat{k} = \max\left\{k \in \{1, \ldots, L\} : \max_{j<k} m_{kj} \leq 7/12\right\}$.

