# OpenReview forum: "Universal Rates for Active Learning"
_NeurIPS.cc/2024/Conference — NeurIPS 2024 poster_

### Official Review · Reviewer_cBhF · 2024-07-12

**Soundness:** 4
**Presentation:** 4
**Contribution:** 4
**Rating:** 8
**Confidence:** 3

**Summary:**

This work provides a characterization of distribution-dependent learning rates for realizable active learning in terms of combinatorial complexity measures on the hypothesis class -- the main result is that any hypothesis class is universally learnable (i.e. learnable with rate $CR(cn)$ for distribution dependent constants $C$ and $c$) with rate either arbitrarily small, exponentially, linear, or arbitrarily large, depending on a complexity measure profile of the class.

**Strengths:**

This work presents an important contribution to a recent line of work on universal rates which includes similar characterizations in the passive and interactive learning settings. It naturally generates comparisons to the "rate partitions" in the passive and interactive settings given in prior work. The authors introduce a new complexity measure which serves to distinguish hypothesis classes for which the more powerful interactive querying model admits faster universal rates than the standard active learning model, as well as to distinguish hypothesis classes which admit exponential universal rate speed-up over passive learning. Given that the search for settings where active strategies can yield an exponential speedups over passive learning has long been a primary focus in the field of active learning, this seems to me to be a noteworthy contribution.

**Weaknesses:**

Pages 8-9 probably should be polished a bit before publication. I think the content of Appendix A.4, which compares the results to passive and interactive learning analogues of this paper and thus gives clear insight into the power of active queries, probably should take up some real estate in the main body in the next version of the paper.

**Questions:**

The results of this paper state that the universal optimal rate of active learning is exponentially faster than that of passive learning when there does not exist an infinite star tree for $\mathbb{H}$. Do the authors feel that the concept of ``infinite star tree'' yields insight into the phenomenon of active learning algorithms tending not to outperform passive sampling in practice?

**Limitations:**

Yes

---

> ### Author Rebuttal · Authors · 2024-08-05
>
> We would like to thank the Reviewer for their detailed feedback. Please find our answers to your questions below.
>
> > Pages 8-9 probably should be polished a bit before publication. I think the content of Appendix A.4, which compares the results to passive and interactive learning analogues of this paper and thus gives clear insight into the power of active queries, probably should take up some real estate in the main body in the next version of the paper.
>
> Thank you for the suggestions. If the paper gets accepted, we will certainly make use of the extra space to polish the presentation of pages 8-9 and bring the context of Appendix A.4 to the main body.
>
> > The results of this paper state that the universal optimal rate of active learning is exponentially faster than that of passive learning when there does not exist an infinite star tree for $H. Do the authors feel that the concept of ``infinite star tree'' yields insight into the phenomenon of active learning algorithms tending not to outperform passive sampling in practice?
>
> Thank you for your question. Our work aims to understand the fundamental capabilities and limitations of the active learning setting itself.  We do not aim to explain phenomena which may have been observed for specific heuristic active learning methods which have been applied in practice.  This is also an interesting subject, worthy of a separate study.

---

### Official Review · Reviewer_CFsL · 2024-07-13

**Soundness:** 3
**Presentation:** 3
**Contribution:** 3
**Rating:** 7
**Confidence:** 3

**Summary:**

This paper deals with universal active learning of binary classes in the realizable setting, which continues important work in related universal settings (interactive, online, ...). The authors propose a star number based VCL-tree variant, which together with original VCL-trees  characterize the 4 possible rates of universal active learning.

**Strengths:**

A strong paper with deep results further continuing the line of work on universal learning, this time active learning. This work nicely complements the "universal rates of interactive learning" paper where general queries are studied, while here only more common label queries are allowed.

The star number based modification of VCL trees is nice and natural.

**Weaknesses:**

Please address my questions below. Am more than happy to raise my score if my first question on $e^{-n}$ learning of intervals (vs $\Omega(1/\varepsilon)$ rate under the uniform distribution) is clarified.

--- rebuttal update ----
raised from 5 to 7.

**Questions:**

Your example with learning intervals on the real line is indeed very surprising. For example, in the Balcan et al. [2010] paper you cite, the authors discuss that actively learning an interval $[a,b]\subseteq (0,1)$ requires $\Omega(1/\varepsilon)$ queries **under the uniform distirbution on $(0,1)$**. Please clarify this apparent contradiction to your $e^{-n}$ claim. Sorry if I am missing something trivial.

The $o(1/n)$ rates seem somewhat related to the rates achieved by Attias et al., [COLT 2024] for universal regression in the absolute loss case. Are there any connections?

Do high-probability bounds ($\geq 1-\delta$) make sense in this active universal setting? Hanneke and Yang [2015] managed to fully remove the dependence on $\delta$ for uniform active learning

**Limitations:**

See questions.

---

> ### Author Rebuttal · Authors · 2024-08-05
>
> We would like to thank the Reviewer for their detailed feedback. Please find our answers to your questions below.
>
> > Intervals and Balcan et al. [2010].
>
> This is a good point to clarify. Balcan et al. (2010) distinguish between the *true* query complexity and the *verifiable* query complexity of a learning task. The $\Omega(1/\varepsilon)$ query lower for learning intervals under the uniform distribution you refer to holds for the *verifiable* query complexity, not for the *true* query complexity, which is the definition we consider in our work (and in prior works in universal rates).
>
> The *verifiable* query complexity refers to the number of queries needed to both produce an $\varepsilon$-good classifier *and* prove that its error is no more than $\varepsilon$ (with high probability). The *true* query complexity refers to the number of queries needed to only output such an $\varepsilon$-good classifier.
>
> More formally, for a given $\mathcal{H}$ and marginal distribution $\mathcal{D}$ over $\mathcal{X},$ a function $Q(\varepsilon, \delta, h^*)$ is a *verifiable* query complexity if there exists an algorithm $A(n,\delta)$ that outputs a classifier $h_{n,\delta}$ and a value $\hat \varepsilon_{n,\delta} \in (0,1)$ after making at most $n$ label queries, so that for any target labeling function $h^* \in \mathcal{H}, \varepsilon, \delta \in (0,1)$ and for any query budget $n$ it holds that $\Pr[\mathrm{er}(h_{n,\delta}) \leq \hat \varepsilon_{n,\delta}] \geq 1-\delta,$ and for any $n \geq Q(\varepsilon, \delta, h^*)$ it holds that $\Pr[\mathrm{er}(h_{n,\delta}) \leq \hat \varepsilon_{n,\delta} \leq \varepsilon] \geq 1-\delta.$
> The definition of *true* query complexity reads the same way, except now the algorithm is *not* required to output $\hat{\varepsilon}_{n,\delta}$. To illustrate how this subtle difference can have a big impact on the query complexity, consider learning intervals over $(0,1)$ under the uniform marginal distribution, and let us ignore the dependence on $\delta$ for simplicity.
>
> Consider two cases: first assume that the target interval is the empty one. Then, there is a learning algorithm that needs 0 queries to learn a $\varepsilon$-good classifier (in fact, a zero error classifier). On the other hand, if the target interval has width $w > 0$, there is a learning algorithm that learns an $\varepsilon$-good classifier needing $O(1/w + \log(1/\varepsilon)) = O(\log(1/\varepsilon)) $ queries using the following strategy: initially the algorithm queries points uniformly at random until it finds some point $x^*$ that has label +1 (this will take roughly $1/w$ many queries) and then does binary search on the two intervals $[0,x^*], [x^*,1]$ to find two $\varepsilon$-approximate endpoints of the target interval, which requires $O(\log(1/\varepsilon))$ many queries. Thus, overall the strategy to learn an $\varepsilon$-good classifier for all target intervals is the following two phase approach: start querying points uniformly at random until either a point with label 1 appears, or the query budget runs out. In the former case, proceed to the “binary search” phase, otherwise output the all zero classifier. The previous argument shows that the *true* query complexity of this algorithm is $O(\log(1/\varepsilon))$.
> Let us now consider the *verifiable* query complexity. If the target interval is the empty one, then for any given query budget $n$, the previous algorithm (and in fact, any algorithm) can only guarantee that its error is at most $O(1/n)$ (with high probability). However, if we consider its learning curve for all different values of $n$, we will observe an exponentially fast decay no matter what the target interval is, which is exactly what the definition of universal rates is capturing.
>
> For a more formal discussion of the two definitions of sample complexity, we kindly refer you to Definition 1, 2 of Balcan et al. (2010) and the subsequent discussion in their paper.
>
> Please let us know if this clarifies your concern.
>
> > Attias et al. (2024).
>
> Attias et al. [COLT 2024] show that there *exists* a class for which $o(1/n)$ rates are tight in the setting of regression. In our work, we give a *complete characterization* of the classes for which $o(1/n)$ rates are tight in the context of active learning for binary classification. Moreover, their result shows that a class cannot be learned at a rate faster than $o(1/n)$  when it has an infinite (scaled version of the) Littlestone tree, whereas our result shows that a class cannot be learned at a rate faster than $o(1/n)$ when it has an infinite star tree.
> Furthermore, the reason that the $o(1/n)$ rates appear in our work and the $o(1/n)$ rates appear in Attias et al. [COLT 2024] are fundamentally different. In a nutshell, we get $o(1/n)$ because when the class has only finite VCL trees the probability that our algorithm queries the label of an unlabeled point is decreasing as $n \rightarrow \infty$, and we can make correct inferences of the labels of the points we do not query. This allows us to get a set $S$ with  $|S| = \omega(n)$  correctly labeled points, and then train a supervised learning algorithm on these points, which has error $O(1/|S|) = o(1/n)$. In the construction of Attias et al. [COLT 2024], they get $o(1/n)$ rates because the magnitude of the errors they make on unseen points decreases, in expectation, as $n \rightarrow \infty$.
> We will add a discussion comparing the two works in the next version of our paper.
>
> > High-prob. bounds.
>
> Studying high-probability bounds in the universal rates setting is an interesting direction. So far, all the works on universal rates have focused on establishing bounds that hold in expectation, in order to provide a cleaner characterization of the landscape of the optimal learning rates. It is an interesting future direction to see for which regimes the dependence on $\delta$ can be fully removed; for instance, for arbitrarily fast rates this is immediate.

---

> > ### Comment · Reviewer_CFsL · 2024-08-11
> >
> > Thanks for the clarifications! I have raised my score.

---

### Official Review · Reviewer_aq73 · 2024-07-13

**Soundness:** 3
**Presentation:** 3
**Contribution:** 4
**Rating:** 7
**Confidence:** 3

**Summary:**

This paper studies active learning for binary classification. The authors provide a complete characterization of the optimal learning rates for non-adaptive active learning algorithms. The authors also develop an active learning algorithm for partial concept classes with exponential rates.

**Strengths:**

The authors provide a complete characterization of the optimal learning rates for non-adaptive active learning algorithms: arbitrarily fast, exponential, o(1/n), and arbitrarily slow. These results answers an open question by Balcan et al. 2010.

**Weaknesses:**

It seems that the analysis heavily relies on the assumption that the active learning algorithm is non-adaptive. For completeness, can authors provided concrete examples of (i) non-adaptive active learning algorithms, and (ii) methods to convert existing active learning algorithms to its non-adaptive counterpart without performance degradation?

**Questions:**

Besides the comments above, can author provide analysis on the computational aspects of the developed/studied active learning algorithms?

---

> ### Author Rebuttal · Authors · 2024-08-05
>
> We would like to thank the Reviewer for their detailed feedback. Please find our answers to your questions below.
>
> > It seems that the analysis heavily relies on the assumption that the active learning algorithm is non-adaptive. For completeness, can authors provided concrete examples of (i) non-adaptive active learning algorithms, and (ii) methods to convert existing active learning algorithms to its non-adaptive counterpart without performance degradation?
>
> We would like to clarify that the non-adaptivity assumption is on the number of *unlabeled* examples that the algorithm requests. In particular, at the beginning of the execution, the algorithm specifies an arbitrarily large number of unlabeled examples that it will use. For example, this number could be $e^{e^{e^n}}$, where $n$ is the number of label queries it has (or something even larger than that). The label requests that the algorithm makes are indeed adaptive, and can depend on the unlabeled examples as well as the answers to previous label requests. Moreover, the only part of our characterization that relies on that assumption is the lower bound for the $o(1/n)$ rates. We believe that this is a mild assumption to make, since it still allows for the *label requests* to be done *adaptively*, and we place no upper bound whatsoever on the number of unlabeled examples it can request at the beginning of the execution. Furthermore, we believe that a construction similar to the one we use in our lower bound can also be used to handle the case where we do not place any such restrictions on the type of access to the unlabeled examples, but the technical details get more involved.
>
> To give some more concrete examples, the well-known CAL algorithm of Cohn, Atlas, Ladner [1] or the Activized Learning algorithm of Hanneke [2] can be slightly modified to fit within our model by specifying a sufficiently large number of *unlabeled* examples that they need at the beginning of the execution.
>
>
> [1] Cohn, D., Atlas, L. and Ladner, R., 1994. Improving generalization with active learning. Machine learning, 15, pp.201-221.
>
> [2] Hanneke, S., 2012. Activized learning: Transforming passive to active with improved label complexity. The Journal of Machine Learning Research, 13(1), pp.1469-15
>
> > Besides the comments above, can author provide analysis on the computational aspects of the developed/studied active learning algorithms?
>
> This is an interesting question. The computational complexity analysis depends on the type of access we have to the underlying hypothesis class. Our main goal in this work is to come up with approaches that give the optimal rates with respect to the sample complexity of this problem, which is always the first step to understanding the computational complexity. In their current form, our algorithms are not computationally efficient, but we hope and believe that they will inspire computationally efficient approaches that work for, potentially, restricted classes and data-generating distributions.

---

> > ### Comment · Reviewer_aq73 · 2024-08-09
> >
> > I'd like to thank the authors for their explanations.

---

### Decision · Program_Chairs · 2024-09-25

**Decision:**

Accept (poster)

**Comment:**

The reviewers are unanimous that this is an interesting work shedding light on the possible types of learning curves in active learning.